# Trans-lesion synthesis and mismatch repair pathway crosstalk defines chemoresistance and hypermutation mechanisms in glioblastoma

Xing Cheng [1,2,17], Jing An [1,3,17], Jitong Lou [4,5,17], Qisheng Gu [6,7,17], Weimin Ding[8,9], Gaith Nabil Droby [1,10], Yilin Wang [1], Chenghao Wang[4], Yanzhe Gao[1], Jay Ramanlal Anand [1], Abigail Shelton[1], Andrew Benson Satterlee[11], Breanna Mann [11], Yun-Chung Hsiao [12], Chih-Wei Liu [12], Kun Lu[12], Shawn Hingtgen [11], Jiguang Wang [13,14], Zhaoliang Liu [3], C. Ryan Miller [1,15], Di Wu [4,8,16], Cyrus Vaziri [1,8,18] ✉ & Yang Yang [1,8,18] ✉

Almost all Glioblastoma (GBM) are either intrinsically resistant to the chemotherapeutical drug temozolomide (TMZ) or acquire therapy-induced mutations that cause chemoresistance and recurrence. The genome maintenance mechanisms responsible for GBM chemoresistance and hypermutation are unknown. We show that the E3 ubiquitin ligase RAD18 (a proximal regulator of TLS) is activated in a Mismatch repair (MMR)-dependent manner in TMZ-treated GBM cells, promoting post-replicative gap-filling and survival. An unbiased CRISPR screen provides an aerial map of RAD18-interacting DNA damage response (DDR) pathways deployed by GBM to tolerate TMZ genotoxicity. Analysis of mutation signatures from TMZ-treated GBM reveals a role for RAD18 in error-free bypass of $O^6mG$ (the most toxic TMZ-induced lesion), and error-prone bypass of other TMZ-induced lesions. Our analyses of recurrent GBM patient samples establishes a correlation between low *RAD18* expression and hypermutation. Taken together we define molecular underpinnings for the hallmark tumorigenic phenotypes of TMZ-treated GBM.

Glioblastoma (GBM) patients have a dismal prognosis with a median survival of only nine months without treatment, which modestly improves to 15–16 months following standard-of-care surgery and adjuvant chemoradiation[1]. Temozolomide (TMZ) is a DNA-methylating chemotherapeutic agent used in the treatment of brain cancers and is the only FDA-approved first-line chemotherapeutic drug for GBM[2]. TMZ induces $N^7$-methyl guanine ($N^7mG$), 3-methyl adenine ($N^3mA$), and $O^6$-methyl guanine ($O^6mG$) lesions, each contributing ~70%, 25%, and 5% of the total TMZ-induced DNA damage, respectively[3]. In most cells, $N^7mG$ and $N^3mA$ lesions generate apurinic sites that are ultimately repaired by the Base Excision Repair (BER) pathway[4]. Although $O^6mG$ comprise <5% of all TMZ-induced lesions, this species is considered to mediate the toxic and antineoplastic effects of TMZ[5]. Cells expressing the suicide enzyme $O^6$-methylguanine-DNA methyltransferase (MGMT) directly remove $O^6mG$ adducts and are refractory to TMZ-induced toxicity[6,7]. $O^6mG$ does not directly interfere with or perturb DNA synthesis. Instead, it is thought that $O^6mG$ adducts mispair with thymine during the first round of DNA replication following alkylation damage[8]. However, lethal DNA damage and cytotoxicity arise in a second cell cycle due to the action of MMR machinery[9–12].

The MMR pathway detects $O^6$mG:T mismatches and removes the mis-paired thymine residues[13,14]. However, the $O^6$mG lesion persists and MMR-mediated gap-filling reforms the $O^6$mG:T mis-pair resulting in additional futile MMR cycles[15–19]. The precise mechanism by which MMR futile cycling leads to cytotoxicity has yet to be established. A widely favoured model posits that the long single-stranded DNA (ssDNA) gaps generated by futile MMR cycles interfere with DNA replication forks[20,21]. However, the specific DNA repair and DNA replication fork-protective mechanisms that allow GBM to tolerate TMZ-induced lesions remain poorly characterized. Therefore, defining the DNA damage response (DDR) effector pathways that respond to TMZ-induced DNA lesions may reveal new vulnerabilities and opportunities for GBM therapy.

All GBM eventually become TMZ-refractory and recur[22]. It is suggested that therapy-induced mutations account for the adaptations that allow GBM to resist TMZ. Hypermutation (i.e., a high mutation burden) is rare in newly-diagnosed gliomas, but has been described in recurrent tumors from TMZ-treated patients[23–25]. The extent to which TMZ-induced hypermutation causes GBM recurrence is still debated[25]. Nevertheless, it is clear that tumors from TMZ-treated GBM patients harbor a specific mutational signature (designated Single Base Substitution 11 or SBS 11) which is characterized by a preponderance of G:C > A:T transitions at non-CpG sites[26]. MMR pathway dysfunction and *MGMT* gene promoter methylation are associated with hypermutation[24,27–30]. However, the underlying mechanisms that mediate therapy-induced mutation signatures and lead to hypermutation only in a subset of recurrent GBM (rGBM) are not fully understood.

Here we sought to elucidate the DDR mechanisms that allow GBM to tolerate TMZ-induced genotoxicity and mediate therapy-induced mutagenesis. Trans-Lesion Synthesis (TLS) pathway is a DNA damage-tolerant and error-prone mode of DNA replication. TLS relies on specialized error-prone "Y-family" DNA polymerases to replicate damaged templates and fill the post-replicative ssDNA gaps, including gaps arising due to spontaneous DNA replication defects in cancer cells[31–33]. Thus, TLS could provide a potential explanation for two major hallmarks of treatment-refractory GBM, namely DNA damage tolerance and mutability.

Recruitment of the Y-family TLS polymerases to damaged DNA is facilitated by the E3 ubiquitin ligase RAD18[34]. RAD18 is recruited to sites of DNA replication stalling or repair synthesis via direct interactions with Replication Protein A (RPA)-coated ssDNA[35,36]. RAD18 then mono-ubiquitinates the conserved residue K164 of PCNA molecules residing in the vicinity of damaged DNA. Interactions between Y-family polymerases and mono-ubiquitinated PCNA constitute a "TLS polymerase switch" that activates post-replicative repair synthesis[37]. Because RAD18 is a proximal master regulator of TLS polymerases, we tested a role for RAD18 in the response to TMZ-induced genotoxicity. We show that a RAD18-Polκ signaling axis is activated in an MMR-dependent manner in TMZ-treated GBM and contributes to cell survival. Additionally, our unbiased genetic screens reveal the RAD18-interacting DDR network that contributes to tolerance of TMZ-induced DNA damage. Finally, our analysis of genomes from TMZ-treated GBM cells defines the contribution of RAD18 to TMZ-induced mutational scars. Taken together, our work provides a mechanistic view of how intersecting genome maintenance pathways mediate ssDNA gap repair, confer survival and reshape the genome in the important disease setting of GBM.

## Results

### TMZ activates the RAD18-mediated TLS pathway in astrocytes and GBM cell lines

To determine whether the RAD18 pathway is activated in response to TMZ, we measured levels of mono-ubiquitinated PCNA in TMZ-treated astrocyte and GBM lines. As shown in Fig. 1a–c, TMZ induced a robust PCNA mono-ubiquitination response in HRAS-transformed normal human astrocytes (NHARas)[38], and in the MGMT-deficient U87 and U373 GBM cell lines. TMZ treatment redistributed RAD18 to RPA-containing nuclear puncta representing DNA repair foci (Fig. 1d). TMZ-induced PCNA mono-ubiquitination was abrogated in $RAD18^{-/-}$ U373 cells (Fig. 1c). In the absence of $RAD18$, TMZ-treatment led to elevated expression of phospho-CHK1 and phospho-ATM (Fig. 1c), markers of unresolved ssDNA gaps and DNA DSB respectively. Therefore, TMZ-induced DNA damage promotes RAD18-dependent PCNA mono-ubiquitination which averts the formation of persistent DNA gaps and secondary DSB. It is formally possible that the apical kinases ATM and ATR both contribute to TMZ-induced CHK1 phosphorylation. However, RAD18-dependent PCNA mono-ubiquitination is best explained by TMZ-induced accumulation of ssDNA independently of DSB formation, as we have shown previously for other genotoxins such as ultraviolet radiation and benzo[a]pyrene[39,40].

PCNA mono-ubiquitination levels peaked 24–48 h following TMZ treatment (Fig. 1c). The delayed kinetics of PCNA-mono-ubiquitination suggest that TMZ-induced RAD18 activation requires a second round of DNA synthesis. Indeed, the number of cells actively synthesizing DNA was not significantly affected during the first 24 h of TMZ-treatment (Fig. 1e). However, 48 h post-TMZ treatment, we observed accumulation of BrdU-positive cells with late-S/G2 DNA content (Fig. 1e). Supplementary Fig. 1 shows that in nocodazole-synchronized cells PCNA mono-ubiquitination peaks in the second cell cycle following TMZ treatment. Critically, nocodazole treatment to block passage into a second cell cycle abrogated the TMZ-induced PCNA mono-ubiquitination.

In $RAD18$-deficient cells, DNA synthesis rates during late S-phase/G2 were reduced when compared with $RAD18^{+/+}$ U373 cells (Fig. 1e). These results suggest that RAD18 mediates post-replication repair of TMZ-induced DNA damage. In contrast with U373 cells, TMZ treatment induced only a modest level of PCNA mono-ubiquitination in LN18 cells which express MGMT and do not accumulate $O^6$mG (Fig. 1f). However, in LN18 cells co-treated with an MGMT inhibitor, $O^6$-benzylguanine (O6BG), TMZ induced robust PCNA mono-ubiquitination (Fig. 1f). RAD18-loss did not affect steady-state levels of $O^6$mG (Fig. 1g). We conclude that TMZ-induced primary $O^6$mG lesions undergo processing during a second replicative cycle, to generate secondary lesions that are repaired by the RAD18 pathway.

### RAD18 depletion sensitizes GBMs to TMZ

TMZ treatment induced RAD18-dependent PCNA-mono-ubiquitination in both NHA and NHARas cell lines (Fig. 2a). In clonogenic survival assays, RAD18-depletion led to increased TMZ sensitivity in NHARas cells but not in untransformed NHA (Fig. 2b). $RAD18^{-/-}$ derivatives of U373 and U87 GBM cell lines were also TMZ-sensitive when compared with isogenic $RAD18^{+/+}$ parental cells (Fig. 2c, d). In contrast with MGMT-deficient U373 and U87 cells, MGMT-expressing LN18 GBM cells were not sensitized to TMZ by RAD18 ablation (Fig. 2e). However, LN18 cells treated with the MGMT inhibitor O6BG were TMZ-sensitive (Fig. 2e). $RAD18$ knockout further sensitized O6BG-treated cells to TMZ when compared with parental ($RAD18^{+/+}$) LN18 cells. These results are consistent with Fig. 1F in which O6BG-treated $RAD18^{-/-}$ LN18 cells expressed increased levels of pRPA32 (a ssDNA marker), pATM, and γH2AX (DSB markers) when compared with parental LN18 cells. We conclude that RAD18 mediates repair and tolerance of $O^6$mG lesions.

In MGMT-expressing GBM (which do not accumulate $O^6$mG), other TMZ-induced lesions $N^3$-methyl adenine ($N^3$mA) and $N^7$-methyl guanine ($N^7$mG), can be repaired by BER[41]. We measured TMZ sensitivity of parental and $RAD18^{-/-}$ LN18 cells in the presence and absence of methoxyamine (MeOX), a BER inhibitor[42,43]. As shown in Fig. 2f, RAD18 ablation led to TMZ sensitivity of LN18 cells only when we inhibited BER. Therefore, BER and RAD18 are redundant pathways for the repair of $N^3$mA and $N^7$mG lesions in MGMT-expressing cells. As

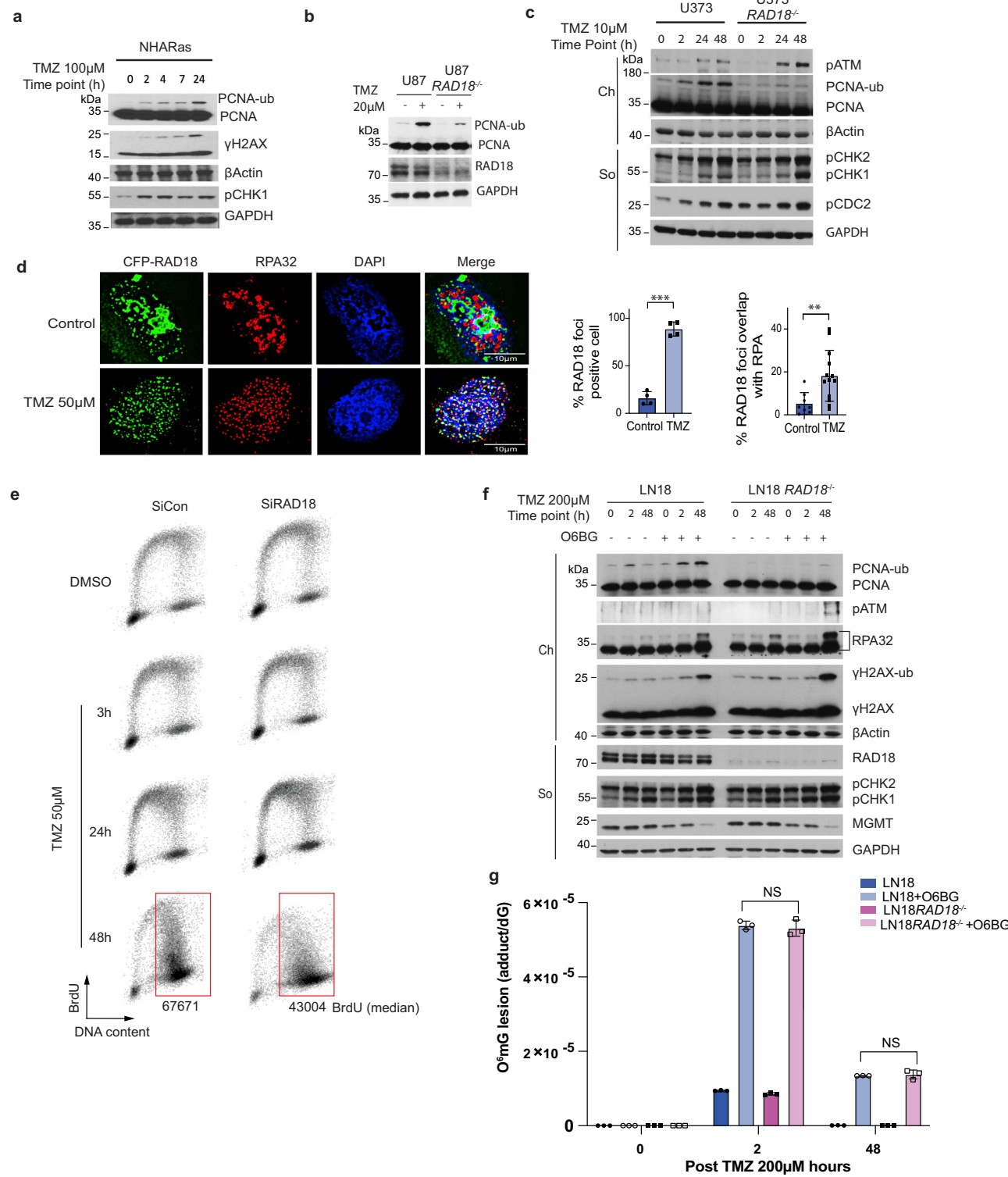

**Fig. 1 | TMZ activates the RAD18 pathway in astrocytes and glioblastoma cells.**
Representative immunoblot showing levels of mono-ubiquitinated PCNA and
indicated DDR markers at different times after TMZ treatment in NHARas (**a**), U87
and U87 *RAD18*$^{-/-}$ cells (**b**), or U373 and U373 *RAD18*$^{-/-}$ cells (**c**). These experiments
had been performed at least three times with similar results. **d** Immunofluorescence
microscopy images showing localization of CFP-RAD18 to nuclear foci in U373 cells
after 48 h treatment with 50 μM TMZ; Green: CFP-RAD18, Red: RPA32 (coating
ssDNA), Blue: DAPI (nucleus). Scale bars, μM. The bar charts show the quantification
of cells with CFP-RAD18 nuclear foci and RPA32-co-localizing nuclear foci. Data
points represent mean ± SD of four independent experiments for RAD18 foci per-
centage (***$p$ = 7.2992e-06) and ≥8 individual nuclei for RAD18 foci overlap with

RPA (**$p$ = 0.0091); **e** FACS analysis showing BrdU incorporation profiles in U373
cells transfected with siCon or siRAD18, followed by 50 μM TMZ treatment for 0, 3,
24, 48 h. BrdU median intensities are shown for gated late S and G2M populations of
the 48 h treatment condition. **f** Representative immunoblots of two independent
experiment showing levels of mono-ubiquitinated PCNA and other DDR markers in
LN18 and LN18 *RAD18*$^{-/-}$ cells after 200 μM TMZ-treatment in the presence or
absence of 20 μM O6BG (MGMT inhibitor). **g** Quantification of primary TMZ-
induced lesions in LN18 and LN18 *RAD18*$^{-/-}$ cells following conditional O6BG
treatment. Data points represent mean ± SD of three biological replicates. $p$ values
were determined by two-sided $t$ test (**d**, **g**). NS no significance. Source data are
provided as a Source Data file.

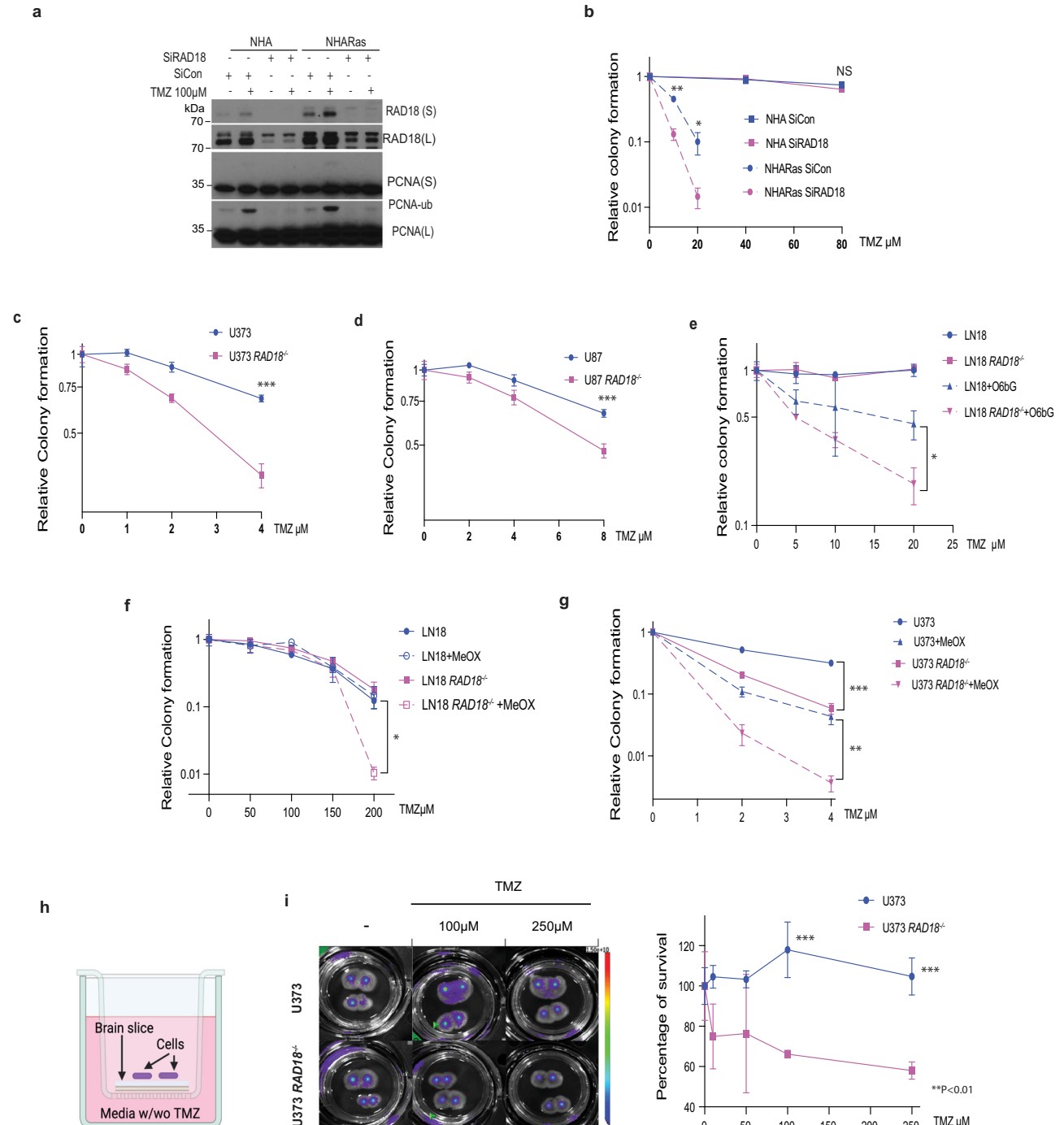

**Fig. 2 | RAD18 promotes TMZ tolerance in Astrocyte and GBM cell lines.**
**a** Representative immunoblot showing the effect of 100 μM TMZ treatment (24 h) on PCNA mono-ubiquitination in NHA and NHARas cells transfected with siCon or siRAD18. **b** Effect of RAD18 knockdown on TMZ sensitivity in NHA and NHARas cells. (**\*\****p* = 0.002, **\****p* = 0.0174). Clonogenic survival assays showing TMZ-sensitivities of parental *(RAD18* $^{+/+}$ *)* and *RAD18* $^{-/-}$ U373 (**c**, **\*\*\****p* = 0.0006) and U87 cells (**d**, **\*\*\****p* = 0.0009). Clonogenic survival assays showing TMZ-sensitivities of *RAD18* $^{+/+}$ and *RAD18* $^{-/-}$ LN18 cells treated with or without 20 μM O6BG (**e**, **\****p* = 0.0128) or 5 mM MeOX (**f**, **\****p* = 0.0111). **g** Clonogenic survival assay TMZ-sensitivities of *RAD18* $^{+/+}$ and *RAD18* $^{-/-}$ U373 cells treated with (**\*\****p* = 0.004) or without (**\*\*\****p* = 1.0460e-05)10 mM MeOX. **h**, **i** TMZ sensitivity of *RAD18* $^{+/+}$ and

*RAD18* $^{-/-}$ U373 cells cultured on viable organo-type brain slice explants. The middle panel shows representative bioluminescence images of firefly luciferase-expressing U373 *RAD18* $^{+/+}$ and RAD18 $^{-/-}$ cells cultured on brain slices and treated with indicated concentrations of TMZ for 4 days. To measure TMZ sensitivity, bioluminescence on day 4 post-TMZ treatment was normalized to day 0 for each treatment group (*n* = 4). 100 μM TMZ **\*\*\****p* = 0.0003, 250 μM TMZ **\*\*\****p* = 9.4745e-05. Data points represent mean ± SD of triplicate determinations; *P* values were determined by two-sided *t* test (**b**–**g**, **i**). Experiment of (**a**–**g**) and (**I**) had been repeated at least three times with similar results. Source data are provided as a Source Data file. (**h**) was created with BioRender.com.

shown in Fig. 2g, RAD18-ablation and BER inhibition had synergistic effects on TMZ sensitivity in U373 cells. Therefore, in MGMT-deficient cells (in which the main cytotoxic lesion is $O^6mG$), BER and RAD18 participate in distinct pathways for tolerance of TMZ-induced lesions. Taken together, the results of Fig. 2f, g identify interactions between RAD18, BER and MGMT-mediated DNA repair pathways, and reveal RAD18 as a vulnerability of both MGMT-deficient and MGMT-expressing GBM. To further validate RAD18 as a dependency of GBM, we determined the TMZ sensitivity of *RAD18*$^{+/+}$ and *RAD18*$^{-/-}$ GBM cells cultured in a newly-developed organotype brain slice culture (OBSC) platform[44]. As shown in Fig. 2h, i, *RAD18*$^{-/-}$ U373 cells cultured on brain slices were highly sensitive to TMZ cytotoxicity when compared with *RAD18*-sufficient parental cells. Taken together, the results of Fig. 2 identify a role for RAD18 in tolerating TMZ-induced cytotoxicity in GBM.

## TMZ-induced RAD18 pathway activation is MMR dependent

48 h after TMZ treatment, we observed reduced BrdU incorporation and an increase in the G2/M-arrested populations of *RAD18*$^{-/-}$ cells relative to *RAD18*$^{+/+}$ control cultures (Figs. 1e and 3a). Flow cytometric analysis of mitosin + phospho-histone H3 doubly-stained nuclei also indicated more robust TMZ-induced G2 arrest in *RAD18*$^{-/-}$ cells when compared with *RAD18*$^{+/+}$ (Fig. 3b, Supplementary Fig. 2). After 48 h of TMZ treatment, RAD18-deficent cells in G2 contained approximately twice as many phospho-RPA (ssDNA marker) and γH2AX (DSB marker) double-positive nuclei when compared with *RAD18*$^{+/+}$ cells (Fig. 3c). Interestingly, our analysis of single-cell RNA (scRNA) gene expression data obtained from fresh primary GBM tumors[45] revealed that *RAD18* expression was higher in S/G2 phase cells compared with cells in G1 (Fig. 3d, Supplementary Fig. 3). This finding is further consistent with an important role for RAD18 in preventing accumulation of post-replicative DNA damage. We conclude that RAD18 prevents accumulation of TMZ-induced post-replicative single-stranded DNA gaps and DSB, averting G2 arrest.

In TMZ-treated cells lacking MGMT, $O^6mG$-thymine mis-pairs are recognized by mismatch repair (MMR) and subsequent futile cycles of DNA repair and thymine reinsertion generate DNA damage[8,46]. As shown in Fig. 3c, TMZ treatment did not increase the ssDNA and DSB double-positive population in *MLH1*$^{-/-}$ cells. RAD18-ablation led to the persistence of DNA damage markers (pCHK1 and pCDC2) in TMZ-treated MMR-sufficient, but not in MMR-deficient (*MLH1*$^{-/-}$) cells (Fig. 3e). MMR-sufficient cells were sensitized to TMZ (as measured by clonogenic survival assays) upon RAD18-depletion (Fig. 3f). However, isogenic cells lacking *MLH1* or *MSH2* were not TMZ-sensitive after RAD18 ablation (Fig. 3f).

TMZ induced robust PCNA ubiquitination in MMR+ cells but not in isogenic cells lacking *MLH1* or *MSH2* (Fig. 3e, g). Therefore, MMR-mediated processing of TMZ-induced lesions is required for RAD18 pathway activation. In MGMT-expressing LN18 cells which do not accumulate $O^6mG$, RAD18-ablation did not lead to persistent DDR signaling after TMZ treatment (Fig. 3h). It is known that following mispair recognition, the nuclease EXO1 excises the daughter strand to expose ssDNA. Ablating EXO1 (using siRNA) led to attenuation of PCNA mono-ubiquitination (Fig. 3i). Taken together, the results of Fig. 3 suggest that TMZ-induced $O^6mG$ lesions are processed by MMR to generate ssDNA gaps that recruit RAD18 to promote PCNA mono-ubiquitination and DNA damage tolerance.

## Polκ mediates TMZ tolerance

To test whether Y-family DNA polymerases mediate TMZ tolerance, we used siRNA to ablate *POLK*, *POLH*, and *POLI* individually, or we used the pharmacological agent JH-RE-06[47] to inhibit REV1 in U373 cells. As shown in Fig. 4b, Polκ-depleted cells (but not cells deficient in Polη, Polι, or REV1) were TMZ-sensitive, suggesting a role for Polκ in repair of TMZ-induced DNA damage. Analysis of TCGA data revealed that low-

grade glioma patients with high-level expression of *RAD18* and *POLK* showed reduced survival probability when compared with *RAD18*-low and *POLK*-low groups respectively (Supplementary Fig. 4). In TMZ-treated cells a GFP-Polκ fusion protein formed nuclear foci in a *RAD18*- and *MLH1*-dependent manner (Fig. 4c). We conclude that MMR-dependent processing of TMZ-induced $O^6mG$ lesions generates secondary DNA damage (likely ssDNA) that activates RAD18, promoting PCNA mono-ubiquitination and Polκ-mediated repair synthesis.

*POLK*$^{-/-}$ U373 cells generated by gene editing partially phenocopied the TMZ sensitivity of *RAD18*$^{-/-}$ cells (Fig. 4d). TMZ sensitivity of *POLK*$^{-/-}$ cells was further exacerbated by RAD18-loss, indicating that Polκ may also have RAD18-independent roles in TMZ-tolerance. Polη inserts C or T across $O^6mG$ lesions[48]. Therefore, we tested a role for Polη as another effector of RAD18-dependent TMZ tolerance. YFP-Polη was redistributed to nuclear repair foci following TMZ-treatment in *RAD18*$^{+/+}$ but not *RAD18*$^{-/-}$ U373 cells (Fig. 4e). Surprisingly, *POLH* knockdown using siRNA led to decreased TMZ sensitivity in both *RAD18*$^{+/+}$ and *RAD18*$^{-/-}$ U373 cells. However, *POLH* siRNA did not affect TMZ sensitivity in *POLK*$^{-/-}$ cells (Fig. 4f). To elucidate the relationship between Polκ and Polη in the TMZ response, we quantified GFP-Polκ foci in *POLH*-depleted cells. As shown in Fig. 4g, *POLH* knockdown led to an increased percentage of cells with PCNA-colocalizing GFP-Polκ foci. We infer that Polη interferes with Polκ PCNA-binding and reduces tolerance of TMZ-induced lesions.

## Defining an integrative network of the DNA damage response to TMZ

To comprehensively define the genome maintenance network (including RAD18-interacting pathways) that mediates tolerance of TMZ-induced DNA damage, we performed a CRISPR-based loss-of-function screen. We transduced *RAD18*$^{+/+}$ and *RAD18*$^{-/-}$ U373 cells with a sgRNA library targeting 504 DDR genes spanning every major DNA repair pathway (Supplementary Data 1). Stably-infected cells were grown for 20 population doublings (PD) in the presence or absence of TMZ. Finally, we sequenced lentivirus integrated cassette DNA to quantify abundance of each sgRNA sequence relative to all mapped reads for each experimental group (Fig. 5a, Supplementary Data 2). Figure 5b plots the normalized counts for all sgRNAs in U373 *RAD18*$^{+/+}$ (blue) and *RAD18*$^{-/-}$ (purple) cells under different experimental conditions. *RAD18* deficiency did not affect the distribution of control non-targeting sgRNAs (NTsgRNA) after 20 PDs in the DMSO-treated cultures. However, we observed a significant reduction in the NTsgRNA count in the *RAD18*$^{-/-}$ line when compared to the *RAD18*$^{+/+}$ cells after 20 PDs in the presence of TMZ treatment. These results indicate that RAD18-loss compromises cell viability under TMZ treatment conditions. Furthermore, we noticed a striking depletion of many DDR-targeting sgRNAs in the TMZ-treated *RAD18*$^{-/-}$ cells when compared with *RAD18*$^{+/+}$. This result indicates a broad general dependency on multiple back-up DNA damage tolerance pathways when RAD18 is absent. The volcano plots in Fig. 5c represent the sigma fold-change (sigmaFC) in Gene Abundance Change Scores vs the statistical significance of depletion or enrichment (Supplementary Data 3). As expected, sgRNAs targeting MMR genes (*MLH1*, *MSH6*, *PMS1*, *PMS2*, and *MSH2*) were highly enriched in TMZ-treated cells, regardless of *RAD18* genotype (Fig. 5c). This positive selection for sgRNAs targeting MMR genes in TMZ-treated cells reflects the role of MMR in generating lethal secondary DNA damage from primary TMZ-induced $O^6mG$ lesions and validates our screen.

To identify the most important mechanisms for TMZ tolerance in *RAD18*$^{+/+}$ and *RAD18*$^{-/-}$ cells, we grouped sgRNAs according to their corresponding DDR pathway(s) and compared relative dropout of the different DDR pathway groups between experimental conditions. The heatmaps in Fig. 5d show the relative dropout of sgRNAs targeting mediators of different branches of the DDR under our screen conditions (Supplementary Data 4 and 5). Interestingly, sgRNAs targeting

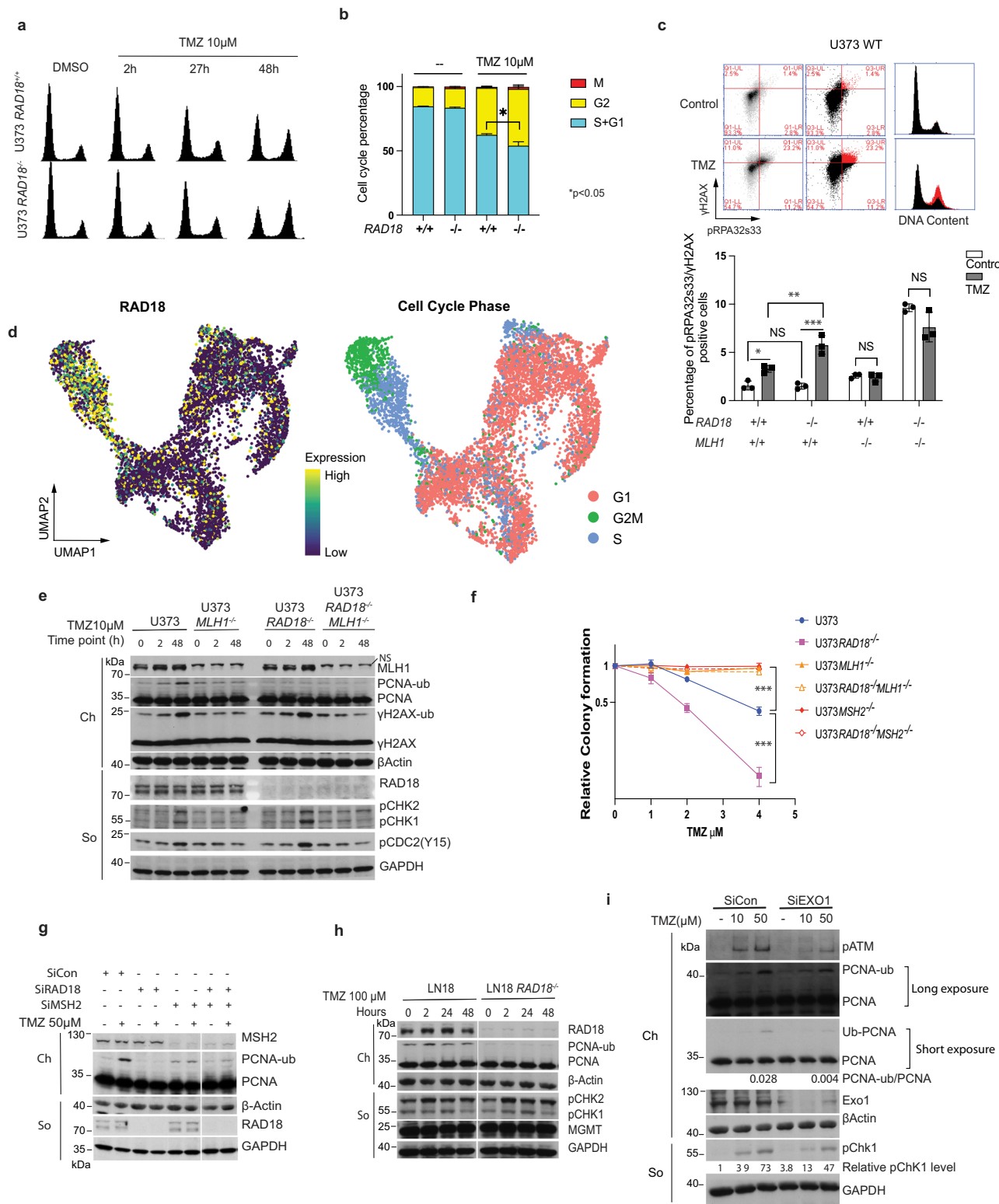

components of the Fanconi Anemia (FA), HR, and checkpoint signaling pathways were preferentially depleted in TMZ-treated cells compared with DMSO-treated control cultures (Fig. 5d), likely indicating that DSB are the ultimate lethal lesions induced by TMZ. Furthermore, FA, NHEJ, checkpoint signaling, and mitosis/spindle assembly checkpoint pathways were preferentially depleted in *RAD18*[−/−] cells, indicating that cells rely on these pathways to process DSB repair when RAD18 is missing (Fig. 5e). This result is consistent with results of Fig. 1 showing a role for RAD18 in suppressing the formation of lethal DSB following

TMZ treatment. Figure 5d, e also indicates minor backup roles for BER, NER, PARP, TS, and nucleotide metabolism in TMZ-treated *RAD18*[−/−] cultures compared with TMZ-treated *RAD18*[+/+] cells.

Based on the significance of drop-out of corresponding sgRNAs (Supplementary Fig. 5A), we selected several DDR factors for independent validation of new roles in TMZ-tolerance. We used pharmacological agents to inhibit CHK2 (PV1019) or DNA-PKs (NU7441) in *RAD18*[+/+] and *RAD18*[−/−] cells. The synergy distribution plots and inhibition scores in Fig. 5e, f show that both CHK2 and DNA-PKs inhibitors

**Fig. 3 | RAD18 promotes tolerance of MMR-dependent DNA damage in TMZ-treated GBM. a** FACS analysis showing cell cycle profiles of *RAD18*⁺/⁺ and *RAD18*⁻/⁻ U373 cells at different times post-treatment with 10 μM TMZ. **b** Cell cycle distribution of *RAD18*⁺/⁺ and *RAD18*⁻/⁻ U373 cells after treatment with 10 μM TMZ for 48 h. Cells in G2 + M or M-phase were enumerated based on staining with mitosin and phospho-histone H3 respectively. *p = 0.0107. **c** FACS analysis of γH2AX and pRPA32s33 in WT, *RAD18*⁻/⁻, *MLH1*⁻/⁻ and *RAD18*⁻/⁻*MLH1*⁻/⁻ U373 cells after a 48 h treatment with TMZ. In the upper panel, the gated cell populations highlighted in red represent the pRPA32s33/γH2AX double-positive cells with 10 μM TMZ in U373 WT cells. The lower panel shows quantification of pRPA32s33/gH2AX double-positive cells in different genotypes after a 48 h treatment with 2 μM TMZ. *p = 0.027, **p = 0.0037, ***p = 0.001, NS, no significance. **d** Uniform Manifold Approximation and Projection (UMAP) scatterplots derived from scRNASeq analysis of GBM cells in patient samples. The UMAP plots show the clustering of individual cells based on RAD18 expression (left) and cell cycle phase (right). **e** Immunoblot showing levels of PCNA mono-ubiquitination and other DDR markers in WT, *RAD18*⁻/⁻, *MLH1*⁻/⁻, and *RAD18*⁻/⁻*MLH1*⁻/⁻ U373 cells treatment with 10 μM TMZ. **f** Clonogenic survival assays showing TMZ-sensitivities of WT, *RAD18*⁻/⁻, *MLH1*⁻/⁻, and *RAD18*⁻/⁻*MLH1*⁻/⁻ U373 cells. Cultures received a single treatment with TMZ daily for 5 days. ***p = 0.001. **g** Representative immunoblot of PCNA mono-ubiquitination in U373 cells transfected with siCon, siRAD18, siMSH2 or siRAD8 + MSH2, and treated conditionally with 50 μM TMZ. **h** Representative immunoblot showing levels of mono-ubiquitinated PCNA and indicated DDR proteins in *RAD18*⁺/⁺ and *Rad18*⁻/⁻ LN18 cells after treatment with 100 μM TMZ for the indicated times. **i** Representative immunoblot showing levels of TMZ-induced PCNA mono-ubiquitination in U373 cells transfected with siCon or siEXO1. The experiments described in (**e**) and (**g**)–(**i**) were performed at least three times with similar results on each occasion. All data points represent mean ± SD of three biological replicates (**b**) and (**c**) and triplicate determinations from a single experiment (representative of three independent experiments) (**f**); *p* values were determined by two-sided *t* test (**b**) and two-sided *t* test with Tukey HSD adjust (**c, f**). Source data are provided as a Source Data file.

synergize with low doses of TMZ to kill *RAD18*⁻/⁻ cells. In *RAD18*⁺/⁺ cells, higher concentrations of TMZ were necessary for synergy with CHK2 or DNA-PKs inhibitors (Fig. 5f, g).

To validate genetic interactions between *RAD18* and the hits from our CRISPR screen, we inactivated *POLD3* in *RAD18*⁺/⁺ and *RAD18*⁻/⁻ U373 cells. As shown in Fig. 5h, combined RAD18 and POLD3 deletion led to greater-than-additive DNA damage sensitivity, validating a synthetic lethal relationship. POLD3 is a component of Polζ and also independently mediates Break Induced Repair (BIR) during Mitotic DNA Synthesis. In contrast with *POLD3*-directed sgRNAs, the sgRNAs targeting other Polζ subunit genes dropped out even in the absence of TMZ-treatment (Fig. 5i). However, sgRNAs against other BIR factors showed a dropout pattern similar to *POLD3*-directed sgRNAs (Fig. 5i). We infer that the role of POLD3 in BIR is important for *RAD18*-deficient GBM to tolerate TMZ. Results of competitive growth assays in Supplementary Fig. 6A, B also demonstrate the effects of deficiencies in FANCD2, PRKDC and POLD3 on TMZ tolerance in U373 cells. Additionally, we validate the interactions of RAD18 with POLD3 and CHK2 using U87 GBM cells (Supplementary Fig. 6C, D). Taken together, we show that GBM cells rely on multiple DNA repair pathways to survive TMZ treatment when RAD18 is missing, thereby illustrating the important central role of RAD18 in tolerance of TMZ-induced DNA damage. We also investigated TMZ induced TLS activity and DNA damage markers in other well-studied GBM cell lines (LN228 MGMT-, D54 MGMT- Supplementary Fig. 7), GBM patient-derived lines (MS21 MGMT-, GBM8 MGMT- Supplementary Fig. 8)[49,50], and Mayo clinic GBM patient-derived xenografts (PDXs) (GBM75 MGMT-, GBM123 MGMT+, GBM12 MGMT-, GBM85 MGMT- provided by Dr. Jann Sarkaria, Mayo Clinic) (Supplementary Fig. 9). In all the MGMT-deficient patient-derived models of cancer (PDMCs) that we investigated, TMZ-tolerance was RAD18-dependent.

## RAD18 mediates both error-prone and error-free bypass of TMZ-induced DNA lesions

We tested the role of *RAD18* in TMZ-induced mutagenesis using U373 cells which readily accumulate O⁶mG (and other TMZ-induced lesions) owing to *MGMT* promoter silencing. Moreover, U373 cells harbor *TP53* mutations and lack growth arrest or apoptotic mechanisms that might otherwise eliminate TMZ-treated cells. Because MMR has also been implicated in hypermutator phenotypes of TMZ-treated gliomas[24,28,30], we also generated *MLH1*⁻/⁻ U373 cell lines in *RAD18*⁺/⁺ and *RAD18*⁻/⁻ backgrounds. The resulting U373 cells with individual or combined deficiencies in *RAD18* and *MLH1* and control parental cells were grown for 20 PDs in the presence of TMZ (or DMSO for controls). We isolated and expanded six clones of U373 cells from each genotype and treatment condition (Fig. 6a). Genomic DNA from each of the clones was used to quantify the mutations that arose de novo during the course of 20 PDs in the presence or absence of TMZ (Supplementary Data 6).

As expected, TMZ treatment significantly increased the mean number of total single nucleotide variation (SNV) arising de novo in *RAD18*⁺/⁺ *MLH1*⁺/⁺ cells (Fig. 6b). In *RAD18*⁻/⁻ *MLH1*⁺/⁺ cells, the mean number of total SNV arising basally exceeded the de novo basal SNV count of *RAD18*⁺/⁺ *MLH1*⁺/⁺ cells. TMZ treatment also led to an increase in the mean SNV count in *RAD18*⁻/⁻ *MLH1*⁺/⁺ cells. However, to our surprise, the mean TMZ-induced increase in total SNV was essentially indistinguishable between *RAD18*⁺/⁺ *MLH1*⁺/⁺ and *RAD18*⁻/⁻ *MLH1*⁺/⁺ cells: TMZ induced 378 SNV in *RAD18*⁻/⁻ *MLH1*⁺/⁺ cells and 396 SNV in *RAD18*⁺/⁺ *MLH1*⁺/⁺ cells.

MMR prevents mutation during DNA replication by removing DNA mismatches from the nascent strand. As expected, all *MLH*⁻/⁻ cells accumulated more SNV de novo over 20 PDs when compared with their *MLH*⁺/⁺ counterparts. However, in an *MLH1*⁻/⁻ background, TMZ only induced a significant number of de novo mutations in *RAD18*⁺/⁺ cells. In *RAD18*⁻/⁻ *MLH1*⁻/⁻ cells there was no significant increase in the number of SNVs following TMZ treatments. We conclude that RAD18-deficiency leads to an increase in basal mutation rates regardless of MMR status. However, RAD18 contributes to TMZ-induced mutations only in cells with active MMR.

We also determined the effect of RAD18 on each of the six possible individual nucleotide substitutions: C>A, C>G, C>T, T>A, T>G, and T>C arising under different experimental conditions. The predominant SNV (C>T) was TMZ-inducible in both *RAD18*⁺/⁺ and *RAD18*⁻/⁻ cells (Fig. 6c, Pattern I). C>T mutations are attributed to error-prone replication of TMZ-induced O⁶mG lesions[26,51]. We therefore infer that RAD18 is not involved in error-prone bypass of TMZ-induced O⁶mG lesions. However, the five less abundant SNVs not attributable to O⁶mG (C>A, C>G, T>A, T>C and T>G) were TMZ-inducible in parental *RAD18*⁺/⁺, but not in *RAD18*⁻/⁻ cells (Fig. 6c, Pattern II). The results of Fig. 6c showing abrogation of TMZ-induced (C>A, C>G, T>A, T>C and T>G) SNVs in *RAD18*⁻/⁻ cells suggest that RAD18 is responsible for error-prone bypass of N⁷mG and N³mA lesions. In MMR-deficient *MLH1*⁻/⁻ cells, the SNV (C>T) was TMZ-inducible in both *RAD18*⁺/⁺ and *RAD18*⁻/⁻ genetic backgrounds (Fig. 6d). However, unlike the *MLH1*⁺/⁺ cells, the other five less abundant SNVs were not induced by TMZ in an *MLH1*⁻/⁻ background (Fig. 6d). Therefore RAD18-mediated error-prone bypass of N⁷mG and N³mA lesions is MMR-dependent.

We also determined the impact of *RAD18* and *MLH1* on TMZ-induced COSMIC "mutation signatures". As expected from previous studies[26], we observed COSMIC signature 11 only in the mutational portraits of TMZ-treated cells (Fig. 6e, Supplementary Data 7). Interestingly, RAD18-deficiency in MMR-sufficient cells led to an ~2-fold increase in signature 11 counts (Fig. 6f), demonstrating that RAD18 promotes error-free repair of a subset of TMZ-induced DNA lesions (i.e., those arising from O⁶mG). However, in an *MLH1*⁻/⁻ background, RAD18-loss did not lead to further increases in Signature 11 counts

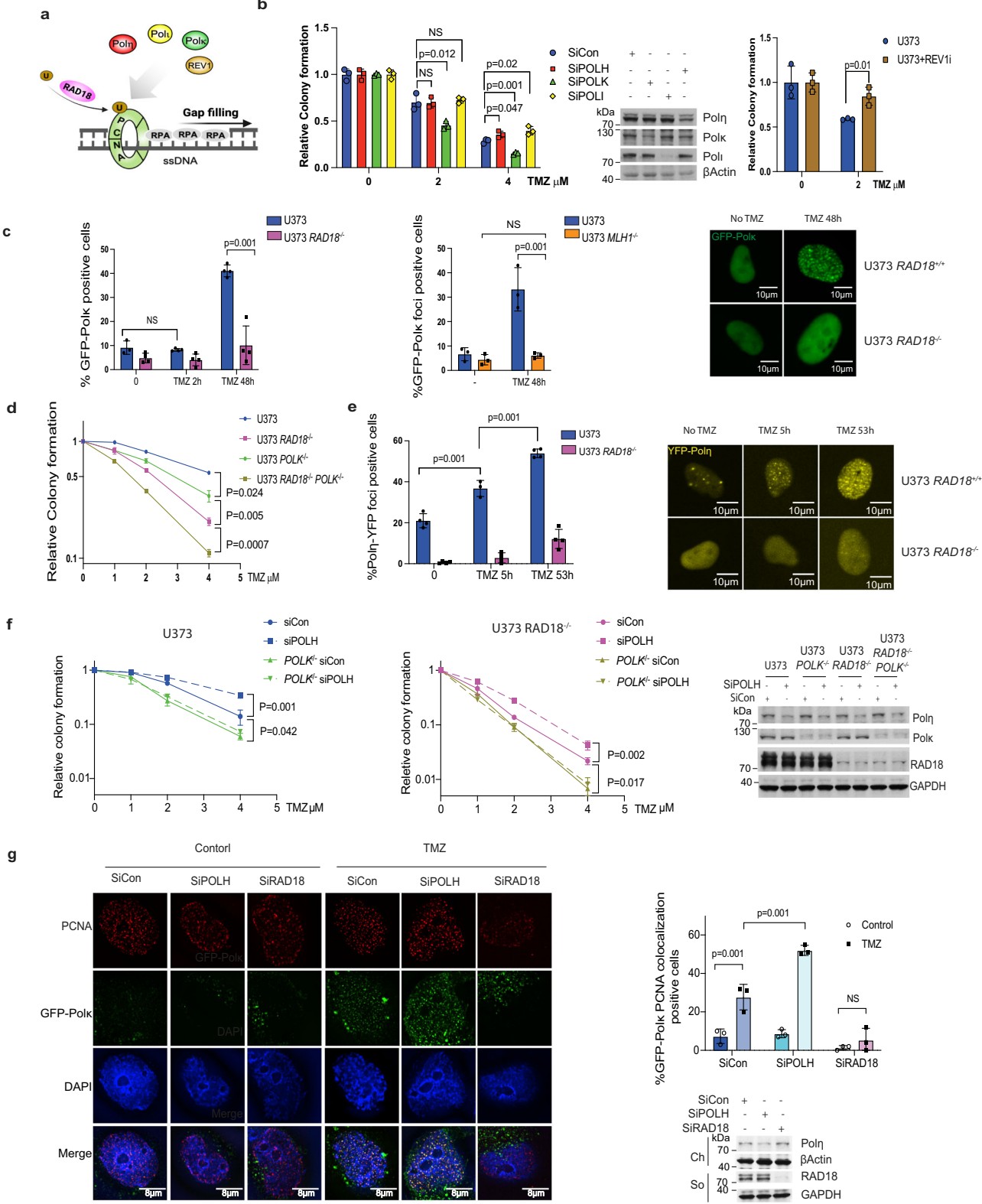

(Fig. 6f). Therefore, RAD18 suppresses mutagenicity of damaged DNA arising via MMR processing of primary TMZ-induced lesions. Taken together, we show that RAD18-mediated DNA damage tolerance and mutagenesis are dissociable in TMZ-treated GBM: RAD18 mediates tolerance of TMZ-induced DNA damage (Figs. 1, 2) and facilitates error-free bypass of the most cytotoxic and mutagenic TMZ-induced lesion ($O^6$mG). However, RAD18 mediates mutagenesis of $N^7$mG and $N^3$mA lesions.

## Low RAD18 expression correlates with hypermutation in recurrent GBM patient

We also investigated correlations between *RAD18* and mutation signature 11 in a patient setting. We extracted and quantified signature 11 counts using genomic sequence data from a multi-institute cohort of TMZ-treated rGBM patients (Supplementary Data 8). *RAD18* expression is correlated with proliferation ($r = 0.75$, $p = 1.7 \times 10^{-14}$, Supplementary Fig. 3D). To exclude the impact of proliferation in our analyses

**Fig. 4 | RAD18-dependent TMZ-tolerance is mediated by Polκ. a** Schematic of TLS DNA polymerase recruited by RAD18 involved in ssDNA gap filling generated by futile MMR cycle. **b** Clonogenic survival assays showing effects of siPOLK, siPOLH, siPOLI, or JH-RE-06 treatments on TMZ sensitivity of U373 cells. The experiment was repeated twice with similar results. **c** Quantification of TMZ-induced GPF-Polκ foci in U373 *RAD18*⁻/⁻ (left) and U373 *MLH1*⁻/⁻ cells (middle) from at least three biological repeats. The right panel shows representative images of GFP-POLK foci in U373 and U373 *RAD18*⁻/⁻ cells treated with or without 20 μM TMZ for 48 h; Scale bars, 10 μm. **d** Clonogenic survival assay showing TMZ-sensitivities of WT, *RAD18*⁻/⁻, *POLK*⁻/⁻ and *RAD18*⁻/⁻*POLK*⁻/⁻ U373 cells. Cells were treated with a single dose of TMZ daily for 5 days. **e** Quantification of TMZ-induced YFP-Polη foci in *RAD18*⁺/⁺ and *RAD18*⁻/⁻ U373 cells from at least 3 biological repeats. The right panels show representative images of cells containing YFP-Polη foci; Scale bars, 10 μm.

**f** Clonogenic survival assays showing effect of *POLK* or *POLH* siRNAs on TMZ sensitivity of *RAD18*⁺/⁺ and *RAD18*⁻/⁻ U373 cells. Cells were treated with the indicated doses of TMZ twice daily for 2 days. **g** Immunofluorescence microscopy images showing the distribution of GFP-Polκ and PCNA in U373 transfected with siCon, siPOLH and siRAD18, followed by 50 μM TMZ treatment for 48 h; Red: PCNA, Green: GFP-Polκ, Blue: (DAPI); Scale bars, 8 μm. The bar chart shows quantification of foci containing co-localized GFP-Polκ and PCNA from three biological repeats. Experiment (**d**) had repeated at least three times and (**f**) had repeat twice with similar results. All data points in (**d,f**) represent the mean ± SD of triplicate determinations; *p* values of (**b**, **d**) were determined by two-sided *t* test, and (**c,e,f,g**) were determined by two-sided *t* test with Tukey HSD adjust. Source data are provided as a Source Data file.

of correlation between *RAD18* levels and signature 11 counts, we adjusted the *RAD18* expression level of each sample for proliferation scores, then separated samples into "high", "medium", and "low" *RAD18* expression groups. Even after controlling for proliferation, hypermutation (sample with >500 counts of Signature 11) was associated with low *RAD18* expression. Consistent with previous reports that *MGMT* promotor methylation (which suppresses MGMT expression) is associated with TMZ-induced hypermutation[24,29,30], low *RAD18* expression was only associated with hypermutability in MGMT-low samples (Fig. 7a). A logistic regression model was further applied and inferred the potential relationship between adjusted *RAD18* expression and hypermutation ($P_{RAD18}$ = 0.014, $P_{MGMT}$ = 0.0436). Our analyses of patient data are consistent with our mutational analyses in cultured cells, and indicate that RAD18 promotes error-free repair of O⁶mG lesions, averting TMZ-induced hypermutations (Fig. 7b).

## Discussion

It is increasingly appreciated that neoplastic cells rely on TLS polymerases for diverse gap-filling functions[52]. For example, Polκ is important for tolerance of ssDNA gaps arising from oncogene-induced DNA replication stress in untransformed mouse and human cells[31]. REV1 is necessary for gap suppression and tolerance of DNA damage from both intrinsic sources and ATR/WEE1-inhibition in ovarian cancer cells[33]. Interestingly, in cisplatin-treated U2OS cells, RAD18, REV1 and Polζ promote gap-filling in G2 while UBC13, RAD51, and REV1-Polζ mediate gap-filling in S-phase[53]. In contrast, we show here that RAD18 and Polκ are crucial for the suppression of TMZ and MMR-induced ssDNA gaps and lethality, while Polι and REV1 are dispensable for DNA damage tolerance in GBM cells. Our finding that Polη-loss and REV1 inhibition promotes TMZ tolerance is likely explained by previous work showing that Polκ and Polη are both RAD18-activated and may compete for recruitment to damaged DNA. The PCNA-Interacting Peptide (PIP) box of Polη binds PCNA with higher affinity than the Polκ PIP motif[54,55]. Therefore, the increased TMZ tolerance of Polη-depleted cells is likely explained by relief of competition for PCNA-binding by Polκ (the preferred polymerase for repair of TMZ-induced ssDNA). Polη and REV1 associate with each other and are epistatic for TLS, likely explaining why REV1 inhibition recapitulates the effects of POLH deficiency. Our study provides paradigms for how ssDNA arises and is remediated by TLS in the unique setting of TMZ-treated GBM. A recent report by Hanisch et al. also showed that HDAC-overexpressing gliomas become TMZ-tolerant owing to transcriptional induction of RAD18[56].

Futile MMR of TMZ-induced O⁶mG-thymine mis-pairs activates ATR[57], signaling a G2 checkpoint that protects from TMZ-induced lethality[58,59]. Here we show coordinate activation of CHK1 and RAD18 in response to TMZ-induced DNA damage, consistent with a common RPA/ssDNA-based activation mechanism for both ATR/CHK1 and RAD18 pathways. Moreover, we show that RAD18-deficiency leads to persistent CHK1 phosphorylation, indicating that RAD18-mediated gap-filling eliminates the TMZ-induced ssDNA tracts that trigger ATR/

CHK1 signaling. Our CRISPR screen reveals a broad aerial view of the DDR landscape and defines networks deployed by GBM to resist TMZ-induced genotoxicity. The sgRNAs targeting FA, HR, and checkpoint signaling pathways were preferentially depleted in TMZ-treated *RAD18*⁺/⁺ cells, likely indicating that DSBs are the ultimate lethal lesions induced by TMZ (Fig. 5d). Through screens with *RAD18*⁻/⁻ GBM, we identify redundant genome maintenance pathways that serve as back-ups for when RAD18 is absent. FA, NHEJ, checkpoint signaling, and mitosis/spindle assembly checkpoint pathways were preferentially depleted in *RAD18*⁻/⁻ cells, indicating that cells rely on these pathways to process DSB repair when RAD18 is missing (Fig. 5e). We also show that the MMR insertion/deletion loop-resolving branch genes *MLH3* and *MSH3* are synthetically-lethal with *RAD18* (Supplementary Fig. 5B). These results indicate that TMZ-induced ssDNA may also be processed to secondary insertion/deletion loop DNA structures in the absence of RAD18-mediated gap-filling (Supplementary Fig. 5C).

The standard-of-care for GBM includes surgical resection, followed by radiation therapy (RT) plus concomitant and maintenance TMZ chemotherapy[60]. However, patient outcomes remain poor due to various intrinsic resistance mechanisms. NHEJ is the predominant pathway for the repair of RT-induced DSBs and confers treatment resistance in GBM[61]. Our CRISPR screen reveals NHEJ as a major backup pathway for tolerance of TMZ-induced DNA damage in *RAD18*⁻/⁻ cells. Pharmacological inhibition of DNA-PK, or knockout of the *PRKDC* gene increases TMZ sensitivity of *RAD18*⁻/⁻ cells. Therefore, further work with orthotopic xenograft models is required to validate our in vitro finding that co-inhibition of TLS and NHEJ pathways might sensitize GBM cells to standard treatments in vivo.

Our finding that RAD18 is important for tolerance of TMZ-induced DNA damage yet has different mutagenic activities on the distinct TMZ-induced lesions (O⁶mG, N⁷mG and N³mA) is highly unexpected and unprecedented. O⁶mG is a highly mutagenic DNA lesion[62]. Both high-fidelity replication polymerases and error-prone Y family polymerases can incorporate T across O⁶mG in vitro[48]. However, the DNA polymerase(s) responsible for C>T mutations at O⁶mG lesions in cells have not been identified. We show that RAD18 suppresses C>T mutations and cosmic Signature 11 in TMZ-treated GBM. We also show that RAD18 and Polκ perform post-replication filling of MMR-dependent ssDNA gaps. Therefore, we tentatively identify Polκ as the RAD18 effector that performs error-free gap-filling of O⁶mG-containing templates, terminating the MMR futile cycle and conferring TMZ-tolerance. Most likely, replicative DNA polymerases (rather than RAD18-dependent TLS polymerases) cause the TMZ-induced C>T mutations that characterize Signature 11. TMZ-induced N⁷mG and N³mA are suspected to be mutagenic[63–65], yet the mechanism of mutagenesis induced by these lesions has been elusive. Our work defines a role for RAD18 in error-prone bypass of N⁷mG and N³mA lesions. Mechanistically, RAD18- and MMR-dependent mutations (T>A, C>A, T>C, C>G and T>G) caused by N⁷mG and N³mA are explained by the results of ref. 66. Those workers demonstrated that DNA replication-independent noncanonical

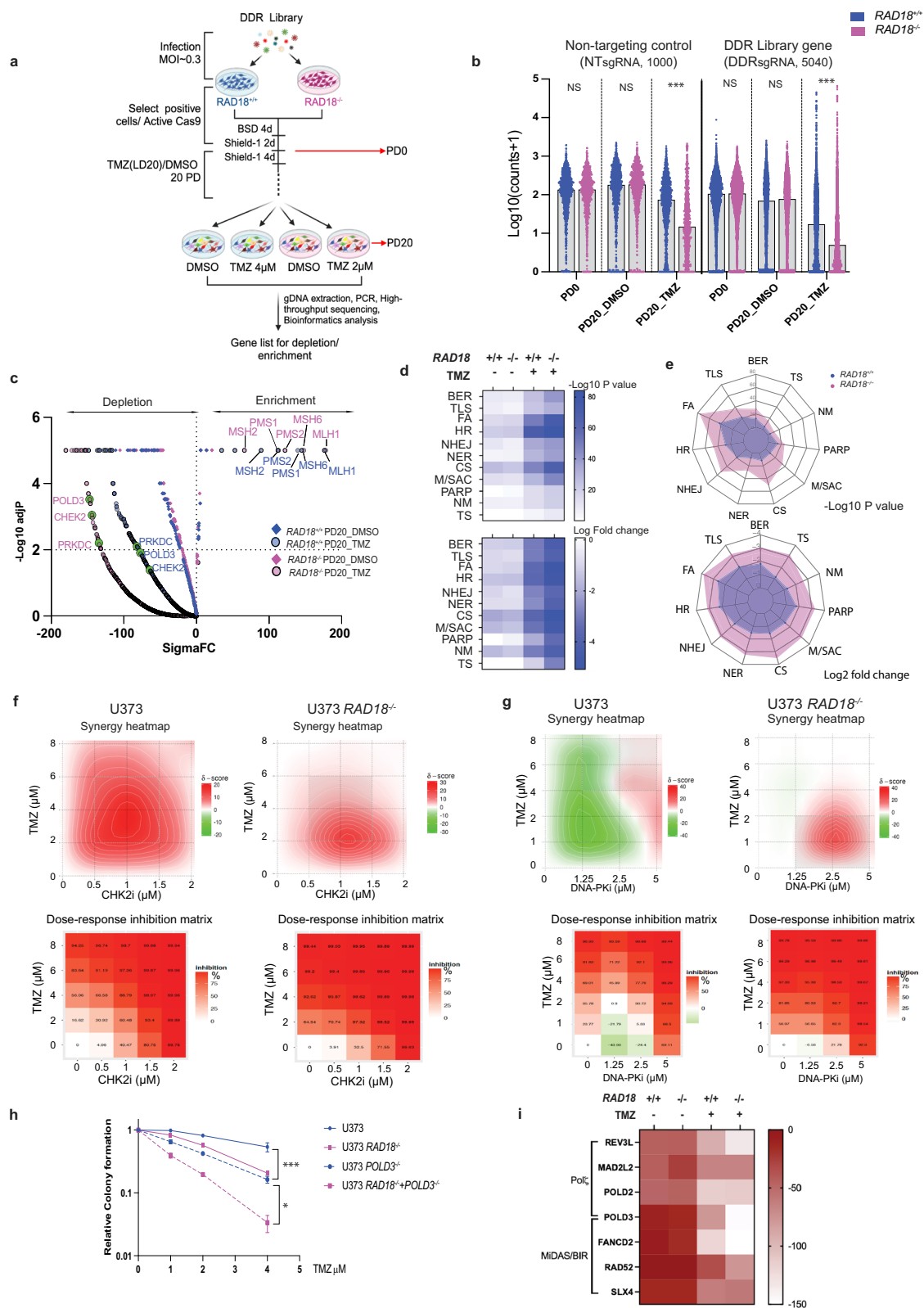

mismatch repair recruits error-prone TLS polymerases to bypass alkylating DNA lesions. It remains to be determined which RAD18-dependent TLS polymerase(s) mediate error-prone replicative bypass of N$^7$mG and N$^3$mA.

A high mutational burden is found in some glioblastomas, yet the underlying mechanisms of hypermutation are not well understood[23–25].

TMZ treatment has been associated with a specific mutational pattern designated COSMIC mutation signature 11[26]. A recent study by Touat et al. identified POLE or MMR mutations in gliomas harboring hypermutations de novo. In the same study an unbiased screen also identified MMR mutations as likely hypermutation drivers following TMZ treatment[30]. Those authors demonstrated that chronic TMZ-treatment

**Fig. 5 | Results of CRISPR screen for RAD18-interacting DDR genes. a** Workflow of genetic screen. MOI Multiplicity of infection, BSD Blasticidin, PD Proliferation doubling, LD Lethal dose. **b** Normalized counts (Log10) of sgRNAs targeting DNA Damage Response (DDR) genes and non-targeting control across indicated samples in *RAD18*^{+/+} (Blue) and *RAD18*^{-/-} (purple) U373 cells. ***p* = 0.001. **c** Volcano plot showing Gene Abundance Change Scores (Sigma FC) vs -Log10 adjusted p-value for sgRNA depletion or enrichment in PD20 groups when compared with PD0. The −Log10 *p* value was calculated using a permutation test for DDR gene-targeting sgRNAs relative to non-targeting control sgRNAs. Black dashed lines indicate thresholds for statistical significance. Enriched sgRNAs targeting MMR genes, and depleted sgRNAs targeting *POLD3*, *CHEK2* and *PRKDC* are highlighted. **d** Heatmap showing relative dropout of sgRNAs grouped by DDR pathway in *RAD18*^{+/+} and *RAD18*^{-/-} cells cultured with or without TMZ for 20 PD. The numbers on the scale indicate −Log10 of paired *t* test *p* value (up) and Log2 fold change (down) of pooled sgRNA counts. BER Base excision repair; TLS Trans-lesion DNA synthesis; FA Fanconi Anemia; HR Homologous recombination; NHEJ Non-homologous end joining;

NER Nucleotide excision repair; CS Checkpoint signaling; M/ASC Mitosis/spindle assembly checkpoint; PARP Poly ADP ribose polymerases; NM Nucleotide metabolism; TS Template switch. **e** Radar plot showing relative dropout (up: *p* value; down: Log2 fold change) of sgRNAs in DDR pathway between TMZ and DMSO control at PD20 in *RAD18*^{+/+} and *RAD18*^{-/-} cells. **f, g** Dose response matrices and synergy heatmaps showing effects of pairwise combinations of TMZ with CHK2i (**e**) or DNA-PKi (**f**) on inhibition of viability in *RAD18*^{+/+} and *RAD18*^{-/-} cells. Mean value of triplicates used to generate heatmaps. **h** Clonogenic survival assays showing TMZ sensitivity of WT, *RAD18*^{-/-}, *POLD3*^{-/-}, and *RAD18*^{-/-}*POLD3*^{-/-} U373 cells. Cultures received a single treatment with TMZ daily for 5 days. All data points represent the mean of triplicates ± SD. ***p* = 0.001, **p* = 0.036 Experiment had been repeated twice with similar results. **i** Heatmap showing sigmaFC of sgRNAs targeting Polζ complex genes and MIDAS/BIR pathway genes. *p* values were determined by two-sided *t* test with Tukey HSD adjust (**b, h**). Source data are provided as a Source Data file. **a** was created with BioRender.com.

of MMR-deficient patient-derived cell lines (but not isogenic MMR-sufficient controls) induced signature 11 hypermutation[30]. Accordingly, those authors concluded that signature 11 was a result of MMR deficiency together with TMZ, rather than a "pure" TMZ signature. Similar to the results of Touat et al., we find that mutation signature 11 is TMZ-inducible. However, in contrast with Touat et al., we show that the acquisition of signature 11 is not contingent upon MMR-deficiency. We note that Touat et al. sequenced genomes from heterogeneous pools of tumor cells, whereas we sequence-analyzed genomes from multiple individual clones. Thus the method employed by Touat and colleagues limited their detection of SNVs to substitutions residing in mutational hotspots. The higher resolution of our platform, which reveals any SNV in a clonal sample, explains why we could detect low levels of SNVs even in MMR+ cells. We conclude that signature 11 is a 'pure' TMZ signature that is suppressed by RAD18. MMR deficiency clearly allows $O^6$mG:T mismatches to persist and become "fixed" as G:C>A:T mutations after a second DNA replication cycle (Fig. 7b). Nevertheless, TMZ -induced hypermutation is probably complex and may not be determined solely by TMZ, MGMT and MMR loss. It is likely that multiple additional genetic factors, possibly including other DNA repair pathways, influence hypermutability.

Clinical studies also indicate that hypermutation in rGBM is a very complicated problem, whose underlying cause and impact on treatment and patient outcomes remain unclear[25,29,67]. Our analysis of genome sequence data from paired primary and recurrent glioma patient samples shows that low-*RAD18* expression is associated with hypermutation and further hints at a role for RAD18 in suppressing TMZ-induced mutations. Inhibiting RAD18-mediated TLS is a potential therapeutic strategy for sensitizing cancer cells to TMZ-induced lethality, yet could also have significant impact on mutagenesis. Therefore, the relationship between RAD18, DNA damage tolerance, the hypermutator phenotype, and clinical outcome must be clarified using in vivo models in order to validate the RAD18 signaling axis as a therapeutic target in glioblastoma.

## Methods
Reagents are available upon request from the corresponding author: C. V.

### Cell lines and gene editing
U373, U87, LN18, LN229, and D54 were purchased from the American Type Culture Collection.

NHA and NHA-RAS cell lines were gifted by Dr. Russell O. Pieper, University of California San Francisco. MS21 (novel low-passage patient-derived GBM line) were generated by Hingtgen lab. All cells were grown at 37 °C in DMEM medium supplemented with 10% fetal bovine serum and penicillin−streptomycin (1%), in humidified chambers with 5% CO2. All cell lines tested negative for mycoplasma

contamination by MycoAlert Mycoplasma Detection Kit. Human STR profiling test (by Labcorp) had used to confirm the identity of U373, U87 and LN18 cells on Jul 22, 2022 (Supplementary Data 10).

GBM8 cells were gifts from H. Wakimoto (Massachusetts General Hospital). GBM8 cells were grown in Neurobasal media (Gibco) with 7.5 ml L-glutamine, 10 ml B27 supplement, 2.5 ml N2 supplement, 1 mg heparin, 10 μg EGF, 10 μg FGF, and 2.5 ml anti-anti.

GBM12, GBM75, GBM85 and GBM123 cells were provided by Dr. Jann Sarkaria (Mayo Clinic, Rochester, MN). GBM75 and GBM123 were cultured in FBS medium, GBM12 and GBM85 were cultured in stem cell medium as the instruction from the provider.

### Organotype brain slice cultures (OBSC)
All OBSCs were generated from P8 Sprague-Dawley rat pups. Dissected brains were mounted on a vibratome platform submerged in ice-cold brain slice media (BSM). Coronal OBSCs were sliced at a thickness of 300 μm at -15 OBSCs/animal. Visibly damaged brains or OBSCs were discarded. Acceptable OBSCs were transferred onto transwell inserts in a six-well culture plate. 1 mL of BSM, as defined by ref. [68] was added under each transwell. The plates were then transferred to a 37 °C incubator with 5% CO2 and 95% humidity. Tumor cells were seeded onto OBSCs 2 h after slicing, with one tumor seeded in the center of each hemisphere for a total of two tumor foci per OBSC. BSM was changed 24 h after slicing. Fluorescence images were taken 24 h post-seeding for normalization of tumor size. Bioluminescence readouts were taken at 96 h post seeding for assessment of cell viability. Luciferin was added underneath the transwell insert and allowed to incubate for 10 min before bioluminescence measurement on an AMI optical imaging system. All work was approved by the Institutional Animal Care and Use Committee at the University of North Carolina-Chapel Hill.

### Immunoblotting
To prepare extracts containing soluble and chromatin-associated proteins, monolayers of cultured cells typically in 10-cm plates were washed twice in ice-cold PBS and lysed in 200−300 μl of ice-cold cytoskeleton buffer (CSK buffer; 10 mM Pipes, pH 6.8, 100 mM NaCl, 300 mM sucrose, 3 mM $MgCl_2$, 1 mM EGTA, 1 mM dithiothreitol, 0.1 mM ATP, 10 mM NaF, and 0.1% Triton X-100) freshly supplemented with protease inhibitor cocktail and PhosSTOP. Lysates were centrifuged at 1500 *g* for 4 min to remove the CSK-insoluble nuclei. The detergent-insoluble nuclear fractions were washed once with 300 μl of CSK buffer and then resuspended in a minimal volume of CSK.

For immunoblotting, cell extracts were separated by SDS-PAGE, transferred to nitrocellulose membranes, and incubated overnight with the primary antibodies and 1 hour with the secondary antibodies in 5% nonfat milk TBST. The information on antibody dilutions/amounts, validation, company names, catalog numbers and clone

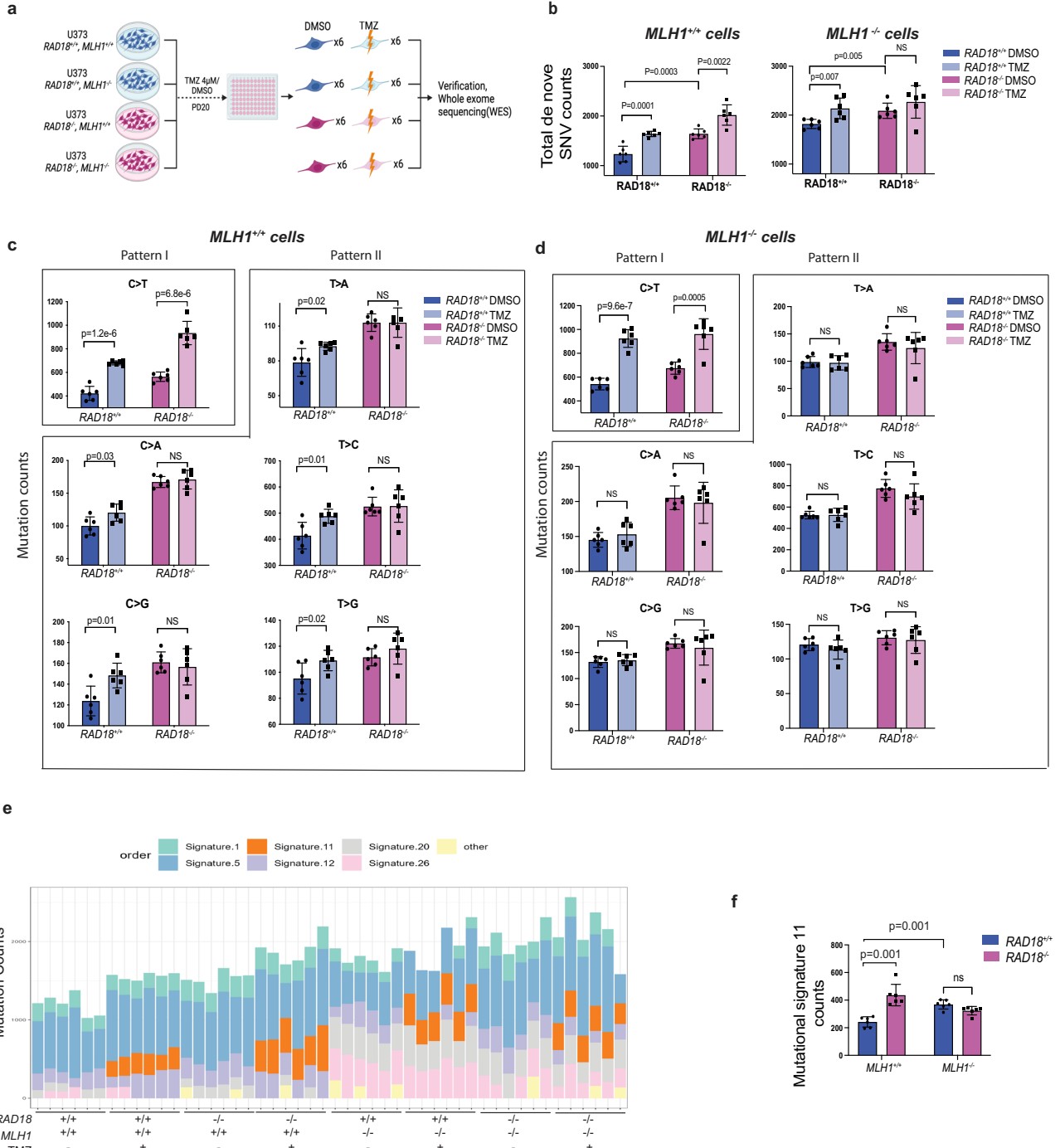

**Fig. 6 | Effect of RAD18 on TMZ-induced mutagenesis. a** Experimental workflow of experiments to define impact of *RAD18* and MMR on TMZ-induced mutagenesis. **b** Bar charts showing numbers of total SNV arising de novo following TMZ treatment over the course of 20 PD in cell lines differing with respect to *RAD18* and MMR status. **c, d** Bar charts showing numbers of each individual SNV arising de novo following TMZ treatment over 20 PD in *MLH1*$^{+/+}$ (**c**) and *MHL1*$^{-/-}$ (**d**) cells. Pattern I (O$^6$mG-induced) and Pattern II (N$^3$mA- and N$^7$mG-induced) SNVs are indicated; NS

no significance. **e** Stacked bar-chart showing contribution of individual COSMIC mutation signatures to the overall mutational patterns of U373 clones obtained from different treatment groups. **f** Bar chart showing effect of *RAD18* and *MLH1* status on mutational Signature 11 counts in clones of TMZ-treated U373 cells. NS, no significance. All data points represent mean ± SD of 6 single clone samples; *p* values were determined by two-sided *t* test (**b, c, d**) and two-sided *t* test with Tukey HSD adjust (**f**). (**a**) was created with BioRender.com.

numbers for monoclonals are included in Supplementary Data 11. Perkin Elmer Western Lightning Plus ECL was used to develop films.

## Plasmids and RNA interference

Plasmid DNA and siRNA oligonucleotides were transfected into U373 cells using electroporation with a GenePulser Xcell (Bio-Rad Laboratories). Electroporation was performed according to the

manufacturer's instructions. 200 µl PBS containing 10$^7$ cells/mL and 1 µM siRNA or 10 µg/ml plasmid DNA was electroporated in a 0.2-cm cuvette using a 150 V, 10 ms, and 1 pulse program. The siRNA oligos were transfected into NHA and NHA-RAS cells using Lipofectamine 2000 (Invitrogen) according to the manufacturer's instructions. Sequences of custom siRNA oligonucleotides used in this study can be found in Supplementary Data 9.

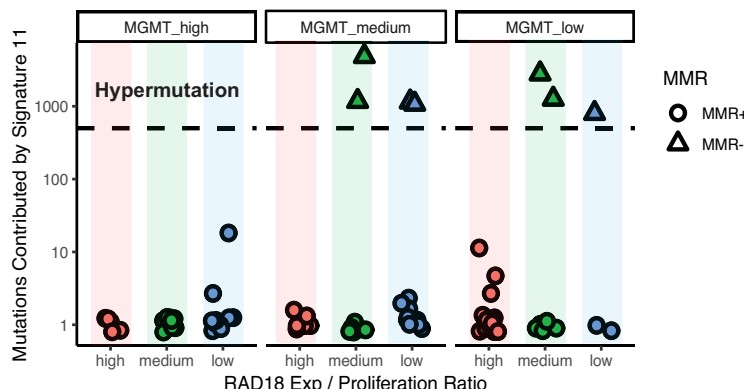

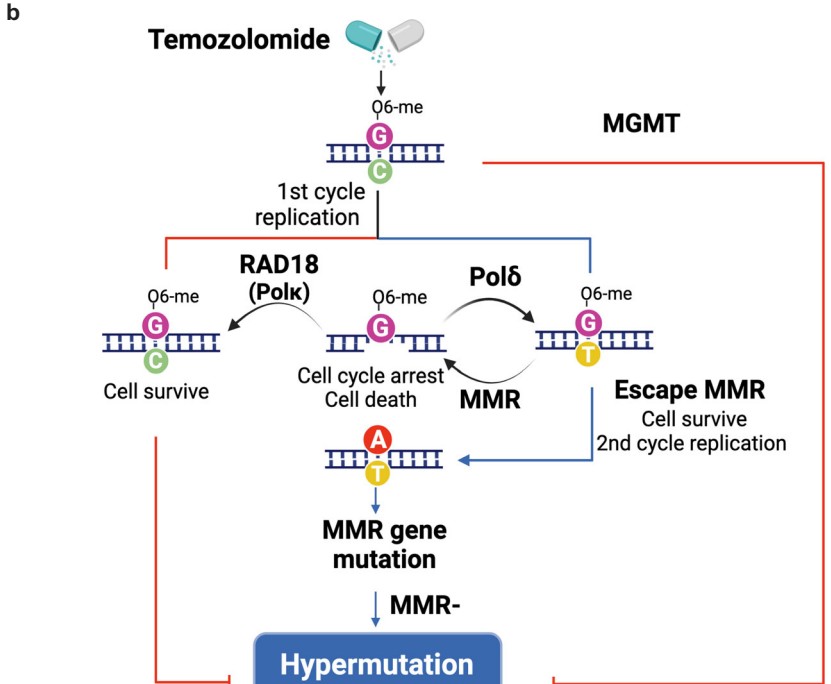

**Fig. 7 | RAD18 suppresses hypermutation in recurrent GBM patients.**
**a** Scatterplot showing the contribution of Signature 11 mutations to individual tumors from a cohort of TMZ-treated recurrent GBM (rGBM) patients. Tumor samples were stratified by *MGMT* status and *RAD18* expression. RAD18 relative expression was corrected for proliferation and was designated high (upper tertile), medium, or low (lower tertile). Based on accepted convention, tumors harboring >500 signature 11 counts (indicated by the black dashed line) were considered "hypermutation". **b** Model describing roles of RAD18, MMR, and MGMT in Hypermutation. (**b**) was created with BioRender.com.

## Adenovirus construction and infection

All adenoviruses were constructed and purified as described previously[69]. In brief, cDNAs encoding RAD18, Polκ or Polη with their tags were subcloned into the pAC.CMV shuttle vector[70]. The resulting shuttle vectors were co-transfected with the pJM17 adenovirus plasmid[70] into 293T cells. Recombinant adenovirus clones were isolated by plaque purification and verified by restriction analysis and Southern blotting. The empty vector AdCon (used as control for adenovirus infections) was derived similarly but by co-transfection of the parental pAC.CMV shuttle vector with pJM17. Adenovirus particles were purified from 293T cell lysates by polyethylene glycol precipitation, CsCl gradient centrifugation, and gel filtration column chromatography. Adenovirus preparations were quantified by $A_{260}$ measurements ($OD_{260}$ 1.0 = ~$10^{12}$ virus particles). Purified adenovirus was stored at −80 °C in small aliquots. U373 cells were infected with $4 \times 10^9$ adenovirus particles/ml by direct addition of purified virus to the culture medium.

## Immunofluorescence microscopy

U373 cells were plated into 4-well chamber slides ($5 \times 10^4$ cells/well). Adenoviruses were added to the cells 24 h before fixing. Cells were washed three times with ice-cold DPBS, then extracted for 5 min in cold CSK buffer on ice surface, washed three times with ice-cold DPBS, then fixed for 20 min in −20 °C methanol. After three washes with DPBS, fixed cells were incubated in blocking buffer (DPBS containing 3% BSA and 5% goat serum) for 1 hour, then incubated with primary antibody (diluted 1:200 in blocking buffer) for 2 h at room temperature. Cells were washed four times with DPBS (5 min/wash) and incubated 1 h in blocking buffer containing secondary antibodies (Alexa Fluor 488 Goat anti-Rabbit and Alexa Fluor 555 Goat anti-mouse, 1:400 dilution).

Secondary antibody was removed by washing four times with DPBS and mounted using Fluoro-gel II containing DAPI (1ug/ml) and a coverslip.

Stained cells were imaged on a Zeiss LSM 710 Spectral Confocal Laser Scanning Microscope (Jena, Germany). We used a Plan-Apochromat 40x/1.40 or 63×/1.40 Oil DIC objective. Excitation lasers were 405 nm (for DAPI), 488 nm (for Alexa Fluor 488), and 543 nm (for Alexa Fluor 555). Emission filter windows were 415–474 (for DAPI), 504–562 (for Alexa Fluor 488), and 504-562 (for Alexa Fluor 555). Over 100 nuclei per condition were scored for quantification.

For the acquisition of representative images, we used an Andor Dragonfly Spinning Disk Confocal Microscope (OXFORD Instruments, England) mounted on a Leica DMi8 microscope stand, equipped with an HC PL APO 100×/1.40 OIL CS2 Leica objective. The pinhole size was set to 40 μm. The camera was a Zyla Plus 4.2MP sCMOS with 2048 × 2048 pixels, with an effective pixel size of 0.063 μm. A piezo-electric Z stage was used to acquire Z stacks at 0.2 μm intervals. Excitation lasers were 405 nm (for DAPI), 488 nm (for Alexa Fluor 488), and 561 nm (for Alexa Fluor 555). Emission filters were 445/46 (for DAPI), 521/38 (for Alexa Fluor 488), and 594/43 (for Alexa Fluor 555). Images were deconvolved in Fusion/Autoquant (version XYZ) using XYZ settings.

For live cell imaging, U373 cells were plated into 4-well glass bottom chamber slides ($5 \times 10^4$ cells/well). TMZ (20 μM) was diluted in culture media and directly added to wells. Adenovirus (AdGFP-Polκ or AdYFP-Pol) was added into culture media 24 hours before imaging. An Andor XD Spinning Disk Confocal Microscope system with Tokai Hit incubator climate control was used for image acquisition. A Hamamatsu Flash4v2 sCMOS camera with 20×/0.75 UPlan S-APO Olympus objective was selected for higher resolution and wider view. FITC and Brightfield channels were selected for acquisition. Over 150 cells per condition were manually scored for nuclear foci using the ImageJ cell counter plug-in.

## Flow cytometry

**BrdU incorporation assay.** Growing cells were incubated with 10 μM BrdU for one hour before trypsinization. Trypsinized cells were suspended in 65% PBS with 35% ethanol and fixed overnight at 4 °C. Fixed cells were denatured using HCl and then neutralized with borax before stained with fluorescent anti-BrdU antibodies for an hour, exactly as described previously[69]. Nuclei were incubated in PBS containing 10 μg/mL of propidium iodide (PI) and 40 μg/mL of RNaseA for 1 h. Stained cells were analyzed using an Accuri C6 plus flow cytometer (BD) and analyzed using the manufacturer's software.

**G2/M cell cycle assay.** Trypsinized cells were resuspended in 1 mL PTEMF buffer (chromatin extraction, fixation and permeabilization, 20 mM PIPES pH 6.8, 10 mM EGTA, 0.2% Triton X-100, 1 mM MgCl2 and 4% formaldehyde) and incubated for 20 min. 5 mL DPBS was added to quench the PTEMF buffer. Fixed cells were resuspended in 1 mL blocking buffer (DPBS + 1%BSA + 0.1% NP-40) and incubated for 1 hour before addition of primary antibodies (Mitosin, 1:200; pH3 1:400 in100 μL blocking buffer) and overnight incubation at 4 °C. Primary antibodies were washed using blocking buffer and cells were incubated in secondary antibodies (Alexa 488 anti-Mouse, 1:400; Alexa 647 anti-Rabbit, 1:400 in 100 μL blocking buffer) for 1 hour at in the dark. After washing in blocking buffer to remove secondary antibodies, cells were re-suspended in blocking buffer containing 10 μg/mL PI and 40 μg/mL RNase A and incubated at 4 °C overnight. Stained cells were analyzed by flow cytometry on an Accuri C6 plus flow cytometer (BD) and analyzed using the manufacturer's software.

**pRPA32s33 and γH2AX staining.** Trypsinized cells were resuspended in 1 mL PTEMF buffer and incubated for 20 min. 5 mL DPBS was added to quench the PTEMF buffer. Fixed cells were resuspended in 1 mL blocking buffer, then incubated for 1 h before adding primary antibodies (pRPA32s33, 1:200; γH2AX 1:400) in 100 μL blocking buffer. After incubating overnight at 4 °C, primary antibodies were removed by washing in blocking buffer. Then, cells were incubated in secondary antibodies (Alexa 488 anti-Mouse, 1:400; Alexa 647 anti-Rabbit, 1:400) in 100 μL blocking buffer for 1 h in the dark. Secondary antibodies were removed by washing in blocking buffer. Then, cells were resuspended in blocking buffer containing 10 μg/mL PI and 40 μg/mL RNase A and incubated at 4 °C overnight. Stained cells were analyzed by flow cytometry on an Accuri C6 plus flow cytometer (BD).

## O⁶mG quantification

LN18 *RAD18*[+/+] and *RAD18*[−/−] cells were pre-treated with 100 mM O6BG for 3 h then co-incubated with 200 μM TMZ + 20 mM O6BG or 200 μM TMZ alone for the indicated times. Genomic DNA was then purified from the cells and used to measure O⁶mG lesions. The experimental procedure and instrument parameters for O⁶mG quantification were similar to those described in previous reports[71–73] with minor modifications. Briefly, the purified DNA samples were spiked with 12.5 fmol [$^{15}$N$_5$]O⁶mG, enzymatically digested with DNAse I (200 units), alkaline phosphatase (5 units), and phosphodiesterase (0.005 units) at 37 °C for 1 h, and fractionated with an off-line HPLC after size-exclusion removal of the digesting enzymes. The purified O⁶mG and [$^{15}$N$_5$]O⁶mG were dried, reconstituted with 14 μL 0.1% formic acid in water, and injected (6 μL per injection) to the nanoLC-ESI-MS/MS system for quantitation of O⁶mG. The quantitating and major product ions of O⁶mG and [$^{15}$N$_5$]O⁶mG were produced by the neutral loss of deoxyribose moiety from each corresponding precursor. Based upon the known spiked internal standard prior to DNA digestion and the signal ratio of O⁶mG: [$^{15}$N$_5$]O⁶mG, the amount of O⁶mG in samples can be estimated, then normalized to the dG content (recorded from the HPLC–UV analysis) in units of adducts/dG.

## Gene editing

Oligonucleotides encoding sgRNAs targeting specific genes or non-targeting control sgRNAs (see Supplemental data "Oligo sequences") were cloned into LentiCRISPRv2 vector[74] which was a gift from Feng Zhang (Addgene plasmid # 52961). The resulting vectors were transformed into Endura Chemically Competent Cells according to the manufacturer's protocol. To generate lentiviruses for gene editing, LentiCRISPRv2-based vectors were mixed with helper plasmids pMD2.G and pSPAX2[75], gifts from Didier Trono (Addgene # 12259 and 12260) using a 4:1:3 ratio (12 μg DNA per transfection reaction) and co-transfected into low-passage HEK 293 T cells using Lipofectamine 2000. Lentivirus-containing culture medium from transfected cells was collected at 24 h and 48 h, filtered through 0.45 μm filter, then aliquoted and stored at −80 °C.

## Viability assays

Cells were seeded at densities of 1000 cells/well (NHA, NHA-RAS), 1500 cells/well (U87, U373), or 2000 cells/well (LN18) in triplicate in six-well plates. All drugs (TMZ, MeOX, CHK2i DNA-PKi) were diluted in growth medium from stock solutions and added directly to the cells. In most experiments, the cell culture medium was replenished with fresh drug every day for 5 days. However, for siRNA knockdown experiments, drugs were replenished twice daily for 2 days beginning 24 h after transfection (thereby ensuring that cells only received the drug at times following knockdown of target mRNAs). For the O6BG co-treatment experiments, LN18 cells were pretreated with 100 μM O6BG 3 h before the first treatment 20 μM O6BG + TMZ. To visualize colonies, culture plates were stained with 0.05% crystal violet in 1× PBS containing 1% methanol and 1% formaldehyde ~12 days after cell seeding. The plates were scanned using an Epson perfection V800 scanner set to "positive film" mode.

## Single-cell RNA-seq data analysis

Smart-seq2 single-cell RNA-seq data generated using malignant GBM cells were obtained from GSM3828672[45]. Data normalization and pre-processing were performed using the Seurat workflow[76]. Batch correction was performed using R package harmony[77]. Cell-cycle scores of individual cells were assigned using Seurat::CellCycleScoring() based on the curated list Seurat::cc.genes.updated.2019 comprising cell cycle markers.

## Defining relationships between RAD18 expression and the TMZ mutation signature using a GBM patient cohort

The integrated GBM cohort containing bulk RNA-seq data, whole exome sequencing data and clinical information was obtained from the published work of ref. 24. Data from *POLE*-mutated samples were excluded from the analysis based on previous work implicating *POLE* alterations in GBM hypermutation[30]. GSVA was performed to score the degree of proliferation based on cell-cycle marker genes from Seurat::cc.genes.updated.2019[78]. MutationalPatterns::fit_to_signatures() was used to reconstruct the mutation matrix and evaluate the mutation counts contributed by Signature 11 based on the COSMIC mutational signatures v2[26]. Linear regression analysis was performed by function lm(RAD18 ~ Proliferation_Score), and difference between RAD18 expression and expected RAD18 expression was taken as adjusted RAD18 expression to reduce the contribution brought by high proportion of proliferating cells. According to the adjusted RAD18 expression level, we divided the patient into "high", "medium" and "low" groups. The logistic regression model on Wang cohort was created using function glm(Hypermutation ~ RAD18_adj + MGMT) with parameter family=binomial(link = "logit"), and p values of variables associated with z values in the model were used to measure the significance of their latent contribution to hypermutation.

## Generating cell lines for CRISPR screen and mutagenesis assay

For the CRISPR-Cas9 screens and mutagenesis assay, we generated isogenic clonal *RAD18*$^{-/-}$ and *RAD18*$^{+/+}$ cell lines by transducing parental U373 cells with a RAD18-directed sgRNA or with a non-targeting sgRNA respectively.

In brief, a LentiCRISPRv2-DD-Cas9-BSD vector (generated in-house, DD: destabilizing domain[79], Supplementary Fig. 10) encoding sgRAD18 or a non-targeting sgRNA was transduced into U373 cells by electroporation. Transfected cells were selected by growth in blasticidin (BSD, 10 μg/mL) for 4 days and then additionally treated with Shield-1 (1 μM) for another 6 days. Single clones of blasticidin-resistant and Shield-1-treated cells were isolated, expanded and tested for *RAD18* disruption by TIDE assay (https://tide.nki.nl/) and western blot. The clonal U373 *RAD18*$^{-/-}$ and *RAD18*$^{+/+}$ cell lines thus obtained were verified lentivirus cassette-free based upon blasticidin-sensitivity.

## CRISPR-Cas9 loss-of-function screen

The DDR-CRISPR lentivirus library comprises 5040 sgRNAs targeting 504 DDR genes (10 independent sgRNAs per gene), and 1000 non-targeting control sgRNAs (Supplementary Data 1). Each domain-focused sgRNA(20nt) is flanked by 5′ and a 3′ universal flanking sequence (Supplementary Fig. 10). The oligonucleotide pool encoding sgRNAs in the library was synthesized by CustomArray Inc. The DDR-CRISPR library was PCR amplified using 017_ArrayF and 018_ArrayR primers (Supplementary Data 9) with Q5 High-Fidelity polymerase. The amplified library DNA was purified using a MinElute PCR purification kit and was cloned into LentiCRISPRv2-DD-Cas9-BSD via NEBuilder HiFi DNA Assembly. The resulting HiFi assembly mixture was transduced into Endura Electrocompetent Cells using a Gene Pulser Xcell electroporator in 1 mm cuvette with 1800 V, 10 μF, 600 Ohm program. Six electroporation reactions were performed to ensure high coverage of the entire library. After a 1 h incubation at 37 °C, transformed bacteria were plated on twelve 15 cm LB ampicillin plates and incubated overnight at 37 °C. Next, 10 mL of LB was added to each plate and the transformed bacterial colonies were recovered using sterile scrapers. Pooled bacterial colonies from three plates were transferred to 500 mL LB and grown with shaking at 37 °C for 3 h. The resulting bacterial cultures were collected and DDR-CRISPR library plasmid DNA was purified using a Qiagen plasmid DNA maxiprep kit.

**Lentivirus generation.** Low-passage HEK 293T cells in 150 mm dishes were grown to 70–90% confluence and transfected using a PEI-based method. Briefly, for each transfection reaction purified plasmid DNAs (4 μg pMD2.G, 6 μg pSPAX2 and 8 μg Library plasmid) were diluted in 1 mL serum-free DMEM and mixed gently. Then, 80 μL PEI was added to the DNA-DMEM solution, mixed gently, and incubated for 20 minutes at room temperature. The resulting DNA-DMEM-PEI solution was added dropwise to 293T cells, which were then returned to the incubator overnight. The next day, transfection medium was removed and replaced with 18 mL fresh complete medium. Virus-containing cell culture medium was collected 24 h later. Cells were given another 18 mL of fresh medium and the virus was collected again 48 h post-transfection. Virus-containing medium was filtered through a 0.45 μm filter, then aliquoted and stored at −80 °C.

**CRISPR-Cas9 screens with DDR CRISPR library.** The DDR CRISPR Library lentivirus was transduced into U373 *RAD18*$^{+/+}$ or U373 *RAD18*$^{-/-}$ cells, at an MOI of ~0.3. 48 h after lentiviral infection, stably-transduced cells were selected in blasticidin (BSD, 10 μg/mL) for 4 days followed by 2 days of treatment with shield-1 (1 μM), the small molecule used to stabilize the destabilizing domain (DD) of Cas9 protein. At this time point, 2 million cells were collected and frozen for control "PD0" samples. Remaining cells were then replated and cultured in the presence of TMZ or with DMSO (for the control group). U373 *RAD18*$^{+/+}$ and U373 *RAD18*$^{-/-}$ cells were treated daily with 4 μM and 2 μM TMZ respectively (doses corresponding to the IC$_{20}$ for each cell line). Shield-1 was maintained in the growth medium for 4 days after the PD0 time point. Every 2–3 days cells were passaged by reseeding $1.6 \times 10^6$ cells into two 15 cm plates (thereby maintaining >250-fold representation of each sgRNA in the library). Using this culture regimen, U373 *RAD18*$^{+/+}$ and U373 *RAD18*$^{-/-}$ cells were grown in the presence or absence of TMZ for 20 population doublings (PD20). Three independent screens were performed. Genomic DNA was extracted from all PD0 and PD20 samples and a two-step PCR was performed to amplify sgRNAs. In the first PCR step, 5 μg gDNA per experimental group (U373 *RAD18*$^{+/+}$ ± TMZ, U373 *RAD18*$^{-/-}$ ± TMZ) was used as a template to achieve ~120× coverage of the DDR CRISPR library. Five 50 μL PCR reactions were performed for each experimental group and each reaction contained 1 μg gDNA, NEB Q5 Hot Start polymerase (added as a 2× master mix), with primer 1st_lib_F_mix and 1st_lib_R (Supplementary Data 9). The product was purified using AMPure XP beads. A second PCR step was used to add indexed sequencing adapters. Each secondary PCR reaction contained 1 ng of purified PCR product from the first PCR step as a template in a 50 μL reaction volume containing indexing primers (Supplementary Data 9). After purification using AMPure XP beads, the 320 bp products were quantified using Qubit and diluted to 1 ng/μL, then sequenced using HiSeq 4000, PE 2×150 (Novogene Co).

**Analysis of sequencing data.** FASTQ sequence files were analyzed with PinAPL-Py (http://pinapl-py.ucsd.edu/)[80]. Briefly, sgRNA counts of PD20 samples relative to PD0 were compared for all experimental groups to determine sgRNA abundance changes that occurred over 20 population doublings. The SUMLFC (the sum of log fold-changes of all sgRNAs targeting the same gene, SigmaFC) algorithm was selected for gene ranking. The DDR pathways heatmap in Fig. 5d was plotted by grouping the sgRNAs according to their corresponding DDR pathways (Supplementary Data 4). Log10 was applied to sgRNA counts to

convert data sets to normal distribution. A paired two-samples $t$ test $p$ value representing the dropout level of PD20 vs PD0 log10 (counts). $-\log10$ ($p$ value) was used to generate the heatmap (Supplementary Data 5). We used the fmsb (version 0.7.3) package in R (version 4.1.2) to make the radar chart. For each cell line and each pathway, we individually performed a two-sided Z-test on the null hypothesis that the genes in the pathway are not on average different in dropouts. The test statistic was calculated as follows. First, using the normalized read counts from PinAPL-Py (Supplementary Data 2), we obtained the ratios between PD20 and PD0 for each guide and log2-transformed them. Then for each gene $g$, we constructed a two-sample Z-statistic comparing TMZ to DMSO as Eq. (1):

$$Z_g = \frac{X_g}{S_g} \tag{1}$$

where $X_g$ is the difference in sample means and $S_g$ is the corresponding standard deviation.

Finally, we combined the gene-specific Z-statistics in each pathway $P$ as Eq. (2):

$$Z_P = \frac{\sum_{g \in P} X_g}{\sqrt{\sum_{g \in P} S_g^2}} \tag{2}$$

which follows a standard normal distribution under the null hypothesis.

## Mutagenesis assay

U373 *RAD18*[+/+] and U373 *RAD18*[−/−] cells were infected with lentiCRISPR virus targeting the *MLH1* gene, or with a lentivirus encoding a non-targeting control sgRNA. Stably transduced cells were selected in puromycin (2 μg/ml). Pools of the resulting U373 cells (*RAD18*[+/+]*MLH1*[+/+], *RAD18*[+/+]*MLH1*[−/−], *RAD18*[−/−]*MLH1*[+/+], and *RAD18*[−/−]*MLH1*[−/−]) were seeded into 10 cm plates ($5 \times 10^5$ cells/plate) and cultured in the presence of 4 μM TMZ (or DMSO for control groups) for 20 population doublings (PD20). Single cells were isolated from each treatment group and expanded into clones that were analyzed via TIDE assay and immunoblot to confirm *RAD18* and *MLH1* genotypes. Expanded cells containing partially-edited clones, or multi-clonal populations were discarded. For each genotype and TMZ treatment condition, six validated clonal cell populations were expanded. Genomic DNA from each clonal line was subject to whole-exome sequencing using NovaSeq 6000 PE150 (Novogene Co).

For all FASTQ sequence files, we firstly discarded reads below the 50 bases long and reads with insufficient base qualities using Trimmomatic 0.39[81]. Next, the trimmed sequencing reads were mapped to the homo sapiens reference genome (assembly GRCh37/hg19) using BWA-MEM alignment algorithm in BWA package (version 0.7.17)[82]. Then we converted the resulting sequence alignment map (SAM) files to binary alignment map (BAM) files using Picard 2.23.4 (http://broadinstitute.github.io/picard). After sorting the BAM files by SAMtools 1.11[83], we removed duplicates and validated the postprocessed BAM files using Picard again.

To call a single nucleotide variant (SNV), a base quality score recalibration was performed by using the tools IndexFeatureFile, BaseRecalibrator, and ApplyBQSR in GATK 4.1.9.0[84]. Known simple nucleotide polymorphisms (dbSNP Build 151) reported in the GRCh37/hg19 background by NCBI (https://ftp.ncbi.nih.gov/snp/organisms/human_9606_b151_GRCh37p13/VCF/common_all_20180423.vcf.gz) were removed as one step in the base quality score recalibration. After the recalibration procedure, we supply the BAM files prepared for substitution-calling to the Mutect2 tool of GATK 4.1.9.0[84] to generate variant calling format (VCF) files. Subsequently, we removed probable technical or germline artifacts by using FilterMutectCalls tool of GATK.

To further reduce false-positive callings, we conducted three additional procedures: 1. we filtered out all INDELs that are greater than 10 bps by using SelectVariants of GATK; 2. using filter tool of SnpSift 4.3t[85], we applied additional filters for mutant allele frequency (≥10%), coverage (≥10×) at particular positions in tumor and normal samples, and number of supporting reads (≥3) for the mutation in the tumor samples; 3. we compare the SNVs and indels to known polymorphisms (U373 MG glioblastoma) using the bcftools in SAMtools. Thus, we obtain the somatic mutations. In addition, we mapped the mutation calls in the filtered VCF files to the protein-coding exon region database (https://earray.chem.agilent.com/suredesign/), and removed the mutations that are not overlapped with any regions in Agilent SureSelect DNA - SureSelect Human All Exon V6 (Supplementary Fig. 11).

## Image quantitation

For clonogenic survival assays, stained plates were scanned and the ImageJ plugin ColonyArea was used to automatically quantify colonies[86]. A threshold of 200 was manually applied to all experiments. Colony survival in treated experimental groups was normalized to untreated controls. For chemical inhibitor studies, synergy distribution heatmaps were generated by SynergyFinder web application (Version 3.0) (https://synergyfinder.fimm.fi/)[87]. To quantify cells containing CFP-RAD18 nuclear foci, and co-localization of GFP-Polκ/PCNA foci, composite images acquired using a Zeiss LSM 710 microscope were generated using Fiji to overlay multiple channels. Cells positive for single foci were counted manually. Cells containing co-localizing foci were identified and enumerated using the cell counter plug-in. To quantify CFP-RAD18/RPA32 percentage overlap, images acquired by an Andor Dragonfly Spinning Disk Confocal Microscope were analyzed using Imaris Microscopy Image Analysis Software (Imaris 9.9). The percentage of surface overlap volume to the surface of CFP-RAD18 channel was used to enumerate the CFP-RAD18/RPA32 overlap. For live cell imaging, the images acquired by Andor XD Spinning Disk Confocal Microscope were opened in Fiji, and foci-positive cells were manually counted using the cell counter plug-in.

## Statistics and reproducibility

The post-hoc Tukey HSD (Honestly Significant Difference) test (https://astatsa.com/OneWay_Anova_with_TukeyHSD/) and multiple unpaired $t$ test (GraphPad Prism10) were used to calculate the $p$ value for experiments with multiple treatment groups. Results are expressed as the mean and standard deviation (SD). The experiments conducted in this study were not randomized, and the investigators were not blinded to allocation during the experiments and outcome assessment. For the patient data analysis, we excluded *POLE*-mutated samples from the analysis based on previous work implicating *POLE* alterations in GBM hypermutation.

## Reporting summary

Further information on research design is available in the Nature Portfolio Reporting Summary linked to this article.

# Data availability

The whole exome sequencing data used in this study are available in the NCBI Sequence Read Archive (SRA) under accession code PRJNA901961. The homo sapiens reference genome GRCh37/hg19 was used for sequence mapping. The protein-coding exon region database (https://earray.chem.agilent.com/suredesign/) and Agilent SureSelect DNA - SureSelect Human All Exon V6 were used to map the mutation calls. Known simple nucleotide polymorphisms (dbSNP Build 151) reported in the GRCh37/hg19 background by NCBI (https://ftp.ncbi.nih.gov/snp/organisms/human_9606_b151_GRCh37p13/VCF/common_all_20180423.vcf.gz) were used to remove the preexisting mutations from analysis. Source data are provided in this paper.

## Code availability

The code used in this study is available at https://doi.org/10.5281/zenodo.10433749[88].

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

## Acknowledgements

The authors thank Dr. Russell O. Pieper from the University of California San Francisco for kindly sharing NHA and NHARas cell lines, Dr. Jann Sarkaria from Mayo Clinic for kindly sharing GBM PDXs, Dr. Hiroaki Wakimoto from Massachusetts General Hospital for kindly sharing the GBM8 cell line, Dr. Pablo Ariel for his assistance in microscopy image acquisition and data analysis. This study was supported by National Institutes of Health grants R01 ES009558, CA215347, and CA229530 (to C.V.), The University of the North Carolina Lineberger Comprehensive Cancer Center Tier 2 Innovation Award (to Y.Y.), The University of North Carolina Translational and Clinical Sciences Institute Award UL1TR001111 (to Y.Y.) and The University of North Carolina Idea seed Grant (to Y.Y.). Microscopy was performed at the Microscopy Services Laboratory at the University of North Carolina, Chapel Hill, which is supported in part by the Cancer Center Core Support grant P30 CA016086 to the UNC Lineberger Comprehensive Cancer Center. The Andor Dragonfly microscope was funded with support from National Institutes of Health grant S10OD030223. The mass spectrometry was supported by the UNC Superfund Research program P42ES031007 and University of North Carolina Center for Environmental Health and Susceptibility grant P30ES010126 (to K.L.). Work in the J.W. lab is supported by grants from Hong Kong RGC (No. CRS_HKUST605/22, 16101021) Hong Kong ITC (No. MHP/004/19, ITCPD/17-9), and Padma Harilela professorship. Work on OBSCs was supported by National Center for Advancing Translational Sciences grant U01TR003715. This project was funded by the UNC Lineberger's  University Cancer Research Fund (UCRF) and the UNC Lineberger Pilot Grants program.

## Author contributions

C.V., Y.Y., and C.R.M. conceived the study. Y.Y., X.C., J.A., J.L., Q.G., W.D., G.D., Y.W., C.W., Y.G., J.R.A, A.S., A.B.S., B.M., Y.H., C.L., J.W., performed experiments. C.V. wrote the Title, Abstract, Introduction, Results, Discussion, Figure legends, cover letter, and responses to reviewers with edits from Y.Y.. Y.Y. prepared the Figures and wrote Materials and Methods with edits from C.V.. J.A. wrote Figure legends with edits from C.V. and Y.Y.. All authors approved the manuscript. C.V., Y.Y, C.R.M., D.W., Z.L., K.L., S.H., and J.W. supervised the work.

## Competing interests

B.M., S.H., and A.B.S. have submitted a provisional patent application based on the OBSC work in this manuscript. The remaining authors declare no competing interests.

## Additional information

[1]Department of Pathology and Laboratory Medicine, University of North Carolina, Chapel Hill, NC 27599, USA. [2]Department of Neuro-Oncology, Chongqing University Cancer Hospital, Chongqing, China. [3]Institute of Cancer Prevention and Treatment, Heilongjiang Academy of Medical Sciences, Harbin Medical University, Harbin, China. [4]Department of Biostatistics, University of North Carolina, Chapel Hill, NC 27599, USA. [5]Eli Lilly and Company, Indianapolis, IN 46285, USA. [6]Shanghai Institute of Immunity and Infection, Chinese Academy of Sciences, Shanghai, China. [7]Department of Immunology, Université Paris Cité, Paris, France. [8]Lineberger Comprehensive Cancer Center, University of North Carolina, Chapel Hill, NC 27599, USA. [9]Oncology Center, Zhujiang Hospital, Southern Medical University, Guangzhou, China. [10]Curriculum in Genetics and Molecular Biology, University of North Carolina, Chapel Hill, NC 27599, USA. [11]Eshelman School of Pharmacy, Division of Pharmacoengineering and Molecular Pharmaceutics, University of North Carolina, Chapel Hill, NC 27599, USA. [12]Department of Environmental Sciences and Engineering, University of North Carolina at Chapel Hill, Chapel Hill, NC 27599, USA. [13]Division of Life Science, Department of Chemical and Biological Engineering, State Key Laboratory of Molecular Neuroscience, The Hong Kong University of Science and Technology, Hong Kong SAR, China. [14]Hong Kong Center for Neurodegenerative Diseases, InnoHK, Hong Kong SAR, China. [15]Department of Pathology, Division of Neuropathology, Heersink School of Medicine, University of Alabama at Birmingham, Birmingham, AL, USA. [16]Division of Oral and Craniofacial Health Science, Adams School of Dentistry, University of North Carolina at Chapel Hill, Chapel Hill, NC, USA. [17]These authors contributed equally: Xing Cheng, Jing An, Jitong Lou, Qisheng Gu. [18]These authors jointly supervised this work: Cyrus Vaziri, Yang Yang. ✉e-mail: cyrus_vaziri@med.unc.edu; yangyang@email.unc.edu

