## [Peer Review File · Nature Communications]

Trans-Lesion Synthesis and Mismatch Repair Pathway
Crosstalk Defines Chemoresistance and Hypermutation
Mechanisms in GlioblastomaReviewers' Comments:

Reviewer #1:

Remarks to the Author:

The manuscript from Cheng et al., examines the role of the E3 ubiquitin ligase RAD18 in regulating the response to the alkylating agent temozolomide in glioblastoma cells. This is a relevant topic as TMZ is a common treatment for glioblastoma, though one which most often ultimately results in recurrence and resistance. Understanding the molecular mechanism of the cellular response to TMZ may provide key insights into improving therapeutic approaches. Generally speaking, the results implicating RAD18 in a response to TMZ are thorough and convincing. The coordination with the mismatch repair pathway makes sense with what is known in the field. However, there are still some mechanistic questions that are not very clear from the current manuscript. Added discussion on a number of these points with a small number of additional experiments may lead to an improved manuscript overall.

1. The concentrations of TMZ used throughout the paper vary. This becomes important as the impact of different DNA repair pathways may change depending on the dose. As higher concentrations are used, the impact of base modifications other than O6-mG come into play. It is hard to compare different experiments when the concentration keeps changing.

2. Determining the timing of RAD18 activity is important for understanding the molecular mechanism. The authors conclude that the effects occur during a second cell cycle due to molecular markers peaking between 24-48 hours. However, this is not very precise. Some of these experiments would be much more informative if cells were synchronized and then followed at various time points to monitor the impacts during the cell cycle more closely.

3. PolK is generally considered a low-fidelity polymerase. Would the introduction of additional mismatches re-activate the MMR damage response?

4. The dose of TMZ used in experiments in Figures 5A and 6A are not provided. These doses are important considering the extent of different base modifications may lead to different DNA damage responses.

5. The conclusions from experiments in which long-term treatments with TMZ were performed are not entirely clear. It would seem a strong selection bias would occur among those cells that survive the treatment, particularly in the MMR proficient cells. Thus, how do the authors rule out that none of the other pathways that impact TMZ survival aren't affected in these surviving clones in Figure 6 and how would their status impact mutation rates?

6. Can the authors explain the discrepancy between the lack of difference in C>T mutations in TMZ treated and untreated MLH1^{-/-}, RAD18^{-/-} cells in 6B and the clear difference in appearance of COSMIC 11 signatures between these two conditions in 6E?

7. Why would RAD18 mediation of mutagenesis of N7mG and N3mA lesions be MMR dependent? One theory might be that these lesions are encountered upon gap filling of a MMR processed O6meG lesions, though then it is not clear why there wouldn't be some effect of RAD18 on fidelity opposite the original O6meG lesion at the same time. The conclusions from Figure 6 are confusing and need more explanation.

8. Figure 7A suggests that all hypermutated tumors are MMR deficient. It is difficult to tell, because of the color coding, if any of the RAD18-high tumors are MMR deficient. If not, it is hard to conclude that the hypermutator status has to do with anything other than MMR status.

Minor points

- Type-o in the name of MLH1 on line 192

- The keys in Figure 4 C and E are very difficult to see. Likewise, this pattern throughout Figure 6 is difficult. Either make the color coding easier to see in the key or use completely different symbols rather than small and large boxes.

Reviewer #2:

Remarks to the Author:

The authors study the mechanisms for TMZ resistance and therapy-induced mutations to understand the pathways involved in these processes. They focus on RAD18, encoding a E3 ubiquitin ligase that functions in translesion synthesis by acting on PCNA. They show that RAD18 is activated in glioblastoma by a mechanism dependent on active mismatch repair (MMR) to promote gap repair via pol kappa TLS and survival to TMZ. A CRISPR screen to identify genes involved in TMZ tolerance revealed factors on DDR pathways including NHEJ, MMEJ and HDR. The mutation signatures seen in TMZ treated glioblastoma indicates that TLS prevents mutation and in its absence, gaps are repaired by error-prone mechanisms most likely using pol delta.

These are very interesting results. The understanding a flow of the paper is marred somewhat by the very numerous types of experiments and large numbers of data in this paper, the confusion of which pathways repair which TMZ-induced lesion (it is not all O6mG), and some of the data presentation.

Graphical abstract:

This could be improved by writing in 1st DNA replication cycle and 2nd DNA replication cycle since it is a key part of the study. Why Rad18 becomes activated only in the second cycle of replication is not known. I remain confused over the bottom seesaw image. The larger lettering is to indicate the more predominant event happening in RAD18+ and RAD18-, I believe. One the left, if survival is increased by RAD18+ and mutation is decreased, should the survival seesaw be elevated and the mutation part of the seesaw be lowered? This would be reversed in rad18- where survival is decreased and mutation is increased. It seems the seesaw is backwards.

In the introduction the authors explain the main types of DNA lesions/modifications induced by TMZ treatment. They then state that only O6mG lesions are acted on by MMR and lead to toxicity. The other lesions, N7mG and N3mA are acted on by BER. However, towards the end of the paper the authors suggest that these lesions too can be acted on by some undefined MMR mechanism with RAD18 (lines 163-164, 415-416). This is confusing. In Figure 2G the authors claim that the effects of BER and RAD18 pathways are additive, but it is difficult to discern this with a small figure with a log scale y axis. If each pathway is contributing about 10X to survival, then the "double" mutant seems to be over 100X reduced from WT (U373), which would be synergistic, not additive. If this were the case, this would have different consequences for TMZ sensitivity.

Figure 1 and line 126-7. In RAD18- there is pCHK2 but this is also seen in RAD18+. pATM and pCHK1 are more specific to RAD18-. What does this mean? Presumably there are fewer DSBs made in RAD18+. It is easier to understand if a simple conclusion is stated. What does it take two cycles to activate RAD18? Is this a result of the futile MMR? Does futile MMR go for several cycles in a single G2?

The authors state that MMR must process the TMZ lesions to have RAD18 activation (line 197). Is the activating trigger a persistent gap or something else? Is it directly behind the fork or in G2? Can the maybe stalled replication with a ssgap be a substrate for MUS81, this inducing the toxicity?

Figure 4B and line 207. There is still significant survival in siPOLK. Why? IS this just the probability of

making a DSB?

Lines 216-217. Here the authors look at RAD18- POLK- double mutants. Again, with the small figure and y-axis log scale it is difficult to say that POLK- phenocopies RAD18- because all have different level of survival.

Line 256, FA is not defined.

Line 323. If RAD18+ is involved in error-prone bypass of N7mG and N3mA, is this using pol zeta? If this is MMR independent, what triggers RAD18 here and since all three types of lesions happen after TMZ treatment, once RAD18 is activated could it not act on O6mG?

Line 371-372. Here the authors suddenly start discussing other DNA damage tolerance and binding of polK vs polH to PCNA. This is relevant but should have been introduced at the beginning. And the novel role, undefined of RAD18 in error-prone (not error-free) bypass of N7mG and N3mA (line 414) seems to come out of nowhere with no molecular explanation or hint of the other role.

Reviewer #3:

Remarks to the Author:

In this manuscript, the authors reported the activation RAD18 as part of the TMZ-induced DDR in GBM cells, which mediates the repair and tolerance of O6mG lesions through Y-family DNA polymerase POLK. Moreover, they performed a CRISPR KO screening in both RAD18 wildtype and KO cells and provided a DDR landscape by which GBM cells resist TMZ-induced genotoxicity. Lastly, they determined the impact of RAD18 on TMZ-induced mutagenesis. This is a potentially interesting study with possible significance. Although the novelty is dampened by a recent publication, Hanisch et al, Cell Death & Disease, 2022, which reported the activation and function of RAD18 in TMZ-treated glioma cells, their unbiased CRISPR screening provide a better understanding of DDR networks after TMZ treatment. Overall, the manuscript is well written with clear presentation, and quality of data is acceptable. The conclusions are justified and of interest to the diverse audience of Nature Communications. The authors may consider the following comments to strengthen the manuscript.

1. The authors should confirm their major conclusions in animal models, such as the impact of RAD18-KO on TMZ sensitivity and the synergetic effect of RAD18-KO and POLD3 KO on TMZ sensitivity.
2. GBM cells with different genetic background might rely on difference DNA repair pathway to survive TMZ treatment when RAD18 is missing. In Fig. 5, the authors performed all experiments in only one GBM cell lines, U373. It is recommended that the authors repeat some of the synergetic experiments in some other GBM lines or patient-derived glioma stem-like cells.
3. The activation and function of RAD18 had been reported by Hanisch et al, Cell Death & Disease, 2022. The authors should cite this paper and discuss their study.
4. The authors' conclusion from Fig. 4B and 4H regarding to the effect of POLH knockdown on TMZ sensitivity is contradictory. Is it caused by different TMZ dose? The authors should perform the experiment in Fig. 4B with higher TMZ dose.
5. In Fig 5D and E, in addition to showing $-\log_{10}$ P value, the authors should also show the fold change value. Moreover, all abbreviations should be described in figure legend.
6. In line 279, "Fig. 5H" should be changed to "Fig. 5I".
7. The numbers in Fig 6B and description in line 303-305 is not matched. Is the y-axis of Fig 6B correctly labeled? Does it mean the number of total SNV or the number of read counts of SNV? They should show the number of total SNV rather than read counts.
8. The authors should provide more detailed information of all the patient samples they used for analysis in Fig. 7. Is it recommended to provide a table including the clinical features, genetic background, and previous treatment strategies for each patient.

Dear reviewers,

I would like to thank you for the kind and helpful comments on our manuscript entitled "*Trans-Lesion Synthesis and Mismatch Repair Pathway Crosstalk Defines Chemoresistance and Hypermutation Mechanisms in Glioblastoma*". We are delighted that the work was considered "relevant, thorough and convincing" (Reviewer 1), and "very interesting" (Reviewer 2) with "*conclusions that are justified and of interest to the diverse audience of Nature Communications*" (Reviewer #3).

We agree completely with all the suggestions for improvements. We would like to submit a revised manuscript containing additional information and results of new experiments that address all comments. We feel that the revised manuscript is strengthened by incorporation of advice and we hope the study will now be suitable for publication.

Our point-by-point response to the reviewers' comments is below. We thank you for the thought and effort you generously put into reviews – we appreciate all of your help and advice.

Sincerely,

Cyrus Vaziri & Yang Yang

REVIEWER COMMENTS

Reviewer #1 (Remarks to the Author):

1. The concentrations of TMZ used throughout the paper vary. This becomes important as the impact of different DNA repair pathways may change depending on the dose. As higher concentrations are used, the impact of base modifications other than O6-mG come into play. It is hard to compare different experiments when the concentration keeps changing.

We agree that this issue should be addressed. In the original manuscript, we used both single- and continuous-treatment procedures to compare TMZ-sensitivities between isogenic cells. In such experiments, the IC₅₀ for single-treatment is higher when compared with continuous TMZ-exposures (although the sensitivities resulting from RAD18/POLK defects are revealed when using either treatment regimen). In the revised manuscript we have substituted new data and consistently present results from continuous TMZ-treatment experiments. We would like to note that the MGMT+ cells (which rapidly repair O6-mG lesions) are highly TMZ-resistant and therefore higher TMZ doses were used in those lines by necessity.

2. Determining the timing of RAD18 activity is important for understanding the molecular mechanism. The authors conclude that the effects occur during a second cell cycle due to molecular markers peaking between 24-48 hours. However, this is not very precise. Some of these experiments would be much more informative if cells were synchronized and then followed at various time points to monitor the impacts during the cell cycle more closely.

Again, thank you for the excellent idea. We have performed the synchronization experiment suggested by the reviewer. In Supplementary Fig. 1, we show results from an experiment in which cells were arrested at metaphase using nocodazole, then released into G1 in the presence or absence of TMZ. Our immunoblot analysis show that TMZ-induced PCNA mono-ubiquitination peaks at time points corresponding to the second cell cycle. Critically, addition of nocodazole to block entry into a second cell cycle abrogates the TMZ-induced PCNA mono-ubiquitination. These experiments are described in lines 136-139 of the revised manuscript.

3. PolK is generally considered a low-fidelity polymerase. Would the introduction of additional mismatches re-activate the MMR damage response?

It is very interesting to speculate that error-prone TLS and mis-insertion of the 'incorrect' nucleotide could create mismatches that reactivate MMR. Our existing results only address this issue indirectly: High-fidelity DNA polymerases preferentially mis-insert T opposite O6mG by ~11-fold over the correct nucleotide (C) (PMID: 17179038). In contrast, owing to a spacious active site that accommodates the O6mG-induced DNA distortion, TLS polymerases mediate a relatively error-free bypass of O6mG lesions (PMID: 11027270). From our genome sequencing analyses, RAD18 inhibits TMZ induced C>T mutations (attributed to error-prone replication of TMZ-induced O6mG lesions). Therefore, we infer that RAD18-Polk mediate error-free bypass of O6mG lesions. However, from the mutagenesis results, we don't see the lower counts of other 5 mutations (T>A, C>A, T>C C>G and T>G) in RAD18^{-/-} group than RAD18^{+/+} group, therefore we can draw a conclusion that Polk might mediate error-free gap filling (or that MMR can repair the additional mismatches). Other dedicated experiments would be needed to test the hypothesis that error-prone TLS activates MMR.

4. The dose of TMZ used in experiments in Figures 5A and 6A are not provided. These doses are important considering the extent of different base modifications may lead to different DNA damage responses.

Thank you for pointing out our omission. We apologize and have specified TMZ concentrations used for the experiments described in Fig. 5A and Fig. 6A of our revised manuscript.

5. The conclusions from experiments in which long-term treatments with TMZ were performed are not entirely clear. It would seem a strong selection bias would occur among those cells that survive the treatment, particularly in the MMR proficient cells. Thus, how do the authors rule out that none of the other pathways that impact TMZ survival aren't affected in these surviving clones in Figure 6 and how would their status impact mutation rates?

The reviewer raises a very interesting question: do we know whether surviving cells represent the starting population and have survivors undergone adaptive changes? Mutagenesis assays like ours are routine - this is actually an intriguing question that could be posed for every experiment in which cells were treated with a mutagen prior to scoring mutations. The tacit assumption is generally that there is not a specific selection process, but that life/death of individual cells treated with a sub-lethal dose is a stochastic response.

In our experiments, cells were grown at the IC₃₀ for TMZ (i.e. a relatively low dose that did not kill 70% of the cells) and therefore there was not a stringent selective bias. We consider it most likely that other than random TMZ-induced mutations, the surviving cells did not diverge significantly from the starting population. It would be very interesting to test whether sustained growth in TMZ rewires the transcriptome or any other process. Such studies would be highly relevant to our understanding of acquired chemoresistance. However, the outcome of such studies would in no way impact our conclusions regarding roles of RAD18 and Polk in determining survival and genome stability outcomes.

6. Can the authors explain the discrepancy between the lack of difference in C>T mutations in TMZ treated and untreated MLH1^{-/-}, RAD18^{-/-} cells in 6B and the clear difference in appearance of COSMIC 11 signatures between these two conditions in 6E?

This is a very good question. The answer is that C>T SNVs are common to many COSMIC mutation signatures (not only the TMZ-induced SBS 11). Therefore in Fig. 6B (which scores all SNVs in the genome), the TMZ-inducible C>T are diluted by the bulk C>T arising via all other basal mutagenic processes (replication stress, metabolites, etc). However, the analyses in Fig. 6E computationally isolates the individual COSMIC mutational

signatures. In so doing, we resolve the C>T mutations representing SBS11 from all the other C>T-containing signatures (SBS1, 5 etc) which are more abundant when compared to SBS11.

7. Why would RAD18 mediation of mutagenesis of N7mG and N3mA lesions be MMR dependent? One theory might be that these lesions are encountered upon gap filling of a MMR processed O6meG lesions, though then it is not clear why there wouldn't be some effect of RAD18 on fidelity opposite the original O6meG lesion at the same time. The conclusions from Figure 6 are confusing and need more explanation.

Thank you – we agree RAD18 / MMR-dependent mutagenesis of N7mG and N3mA lesions is an important and novel finding. Fig 6.C shows that in MMR+ (MLH+/+) cells TMZ-induced mutations other than the O6-mG induced C>T (i.e. T>A, C>A, T>C, C>G and T>G) are RAD18-dependent. We can reasonably infer that those RAD18- and MMR-dependent mutations (T>A, C>A, T>C, C>G and T>G) are induced by the other 2 abundant TMZ lesions, N7mG and N3mA (not by O6mG). Mechanistically, this is explained by the results of Pena-Diaz et al. (PMID: 22864113). Those workers demonstrated that DNA replication-independent noncanonical mismatch repair (ncMMR) recruits error-prone TLS polymerases to bypass alkylating DNA lesions. We have included this explanation in the revised manuscript (lines 429-432).

8. Figure 7A suggests that all hypermutated tumors are MMR deficient. It is difficult to tell, because of the color coding, if any of the RAD18-high tumors are MMR deficient. If not, it is hard to conclude that the hypermutator status has to do with anything other than MMR status.

Apologies for the admittedly poor color choice. We provided a new and more-appropriately labelled version of Fig. 7. The results show that hypermutability is associated with MMR-deficiency (exactly as the reviewer notes), but only in those tumors with low or medium levels of RAD18 expression. This correlation is fully consistent with our results, and suggests that high RAD18 expression represses hypermutation.

Minor points

- Type-o in the name of MLH1 on line 192
Thank you for finding this typo – we have corrected it.

- The keys in Figure 4 C and E are very difficult to see. Likewise, this pattern throughout Figure 6 is difficult. Either make the color coding easier to see in the key or use completely different symbols rather than small and large boxes.

Again, thank you for the helpful comment. We have improved Fig. 4 and Fig. 6 as suggested.

Reviewer #2 (Remarks to the Author):

Graphical abstract:

This could be improved by writing in 1st DNA replication cycle and 2nd DNA replication cycle since it is a key part of the study. Why Rad18 becomes activated only in the second cycle of replication is not known. I remain confused over the bottom seesaw image. The larger lettering is to indicate the more predominant event happening in RAD18+ and RAD18-, I believe. One the left, if survival is increased by RAD18+ and mutation is decreased, should the survival seesaw be elevated and the mutation part of the seesaw be lowered? This would be reversed in rad18- where survival is decreased and mutation is increased. It seems the seesaw is backwards.

Thank you for the excellent comment – we agree with all of these points. In the revised graphical abstract we

have indicated the specific replication cycles in which the different processes occur. We have also eliminated the confusing see-saw and substituted clear statements of survival and mutagenesis outcomes for each genotype.

In the introduction the authors explain the main types of DNA lesions/modifications induced by TMZ treatment. They then state that only O6mG lesions are acted on by MMR and lead to toxicity. The. Other lesions, N7mG and N3mA are acted on by BER. However, towards the end of the paper the authors suggest that these lesions too can be acted on by some undefined MMR mechanism with RAD18 (lines 163-164, 415-416). This is confusing.

Thank you for pointing out an important omission (also noticed by Reviewer 1, comment #7). In lines 429-432 of the revised manuscript we explain how the DNA replication-independent noncanonical mismatch repair (ncMMR) recruits error-prone TLS polymerases to bypass alkylating DNA lesions (Pena-Diaz et al., PMID: 22864113). Those results fully explain how RAD18-dependent recruitment of error-prone TLS polymerase(s) leads to mutation at N7mG and N3mA.

In Figure 2G the authors claim that the effects of BER and RAD18 pathways are additive, but it is difficult to discern this with a small figure with a log scale y axis. If each pathway is contributing about 10X to survival, then the “double” mutant seems to be over 100X reduced from WT (U373), which would be synergistic, not additive. If this were the case, this would have different consequences for TMZ sensitivity.

Thank you for noticing our mistake. From our calculations, the co-inhibition of RAD18 and BER indeed has a synergistic effect on TMZ-sensitivity. Specifically, inhibiting RAD18 and BER individually leads to 5.2-fold and 7.3-fold increases in TMZ sensitivity. However, co-ablation of RAD18 and BER leads to an 86-fold sensitivity to TMZ. An additive response would lead to a ~38-fold sensitization.

It is well accepted in the field that TLS is a redundant and non-epistatic backup pathway for BER (PMID: 23408852, 23295675). Therefore, co-inhibition of BER and RAD18 is expected to have a synergistic and synthetic lethal effect, exactly as we observe.

Figure 1 and line 126-7. In RAD18- there is pCHK2 but this is also seen in RAD18+. pATM and pCHK1 are more specific to RAD18-. What does this mean? Presumably there are fewer DSBs made in RAD18+. It is easier to understand if a simple conclusion is stated. What does it take two cycles to activate RAD18? Is this a result of the futile MMR? Does futile MMR go for several cycles in a single G2? The authors state that MMR must process the TMZ lesions to have RAD18 activation (line 197). Is the activating trigger a persistent gap or something else? Is it directly behind the fork or in G2? Can the maybe stalled replication with a ssgap be a substrate for MUS81, this inducing the toxicity?

Thank you for encouraging us to describe these results and state our conclusions more clearly. In the revised manuscript (lines 127-130) we explain that *“In the absence of RAD18, TMZ-treatment led to elevated expression of phospho-ATM and phospho-CHK1 (Fig. 1C), markers of DNA DSB and ssDNA gaps respectively.”* Moreover, we like the reviewer’s suggestion of stating a simple conclusion that: *“TMZ-induced DNA damage promotes RAD18-dependent PCNA mono-ubiquitination and averts formation of persistent secondary DNA gaps and breaks.”*

Regarding the futile cycle hypothesis and the two-cell cycle requirement for MMR-dependent replicative arrest – this model is an accepted paradigm. Because of space restrictions we cited the relevant papers without elaborating at length on the accepted mechanism. Essentially, DNA replication incorporates T across from O6mG during a first cell cycle. O6mG is not a replication fork-stalling lesion (PMID: 18079180) and

therefore does not activate RAD18 during the first cell cycle. However, O6mG:T processing by MMR causes DNA replication defects that activate TLS in a second replication cycle. Our mutagenesis assay results show that RAD18 mediates relatively error-free bypass of O6mG and thus terminates the MMR futile cycle. Without RAD18, multiple futile cycles of MMR likely occur until C is incorporated against O6mG.

As the reviewer suggests, the TMZ-induced co-localization of RAD18 and RPA32 foci (Fig.1 D) indicate that persistent ssDNA gaps trigger RAD18. From our flow cytometry experiments, most ssDNA-containing cells are G2-arrested, indicating that RAD18 activation is not coupled to S-phase.

The authors state that MMR must process the TMZ lesions to have RAD18 activation (line 197). Is the activating trigger a persistent gap or something else? Is it directly behind the fork or in G2? Can the maybe stalled replication with a ssgap be a substrate for MUS81, this inducing the toxicity?

We like the reviewer's excellent hypothesis that an MMR-induced persistent ssDNA gap is the trigger for activating RAD18. Because EXO1 is involved in MMR, we tested the reviewer's hypothesis that this nuclease contributes to formation of the ssDNA tracts that activate RAD18. Consistent with the hypothesis, ablating EXO1 (using siRNA) led to attenuation of PCNA mono-ubiquitination (described in lines 206-208). Therefore, ssDNA gaps generated during MMR likely explain the RAD18 activation mechanism. We also tested the reviewer's suggestion that MUS81 might also be involved in the RAD18 activation process. Interestingly, MUS81-depletion also attenuated PCNA mono-ubiquitination (although we opted to not include those data in the final manuscript). Nevertheless, MUS81-mediated resolution of recombination intermediates following replication collapse may also contribute to RAD18 activation. We thank the reviewer for suggesting these nice experiments which help reveal the mechanism of MMR-dependent RAD18 activation.

Figure 4B and line 207. There is still significant survival in siPOLK. Why? Is this just the probability of making a DSB?

The TMZ concentration (2uM) we used in Fig. 2B corresponds to the IC₃₀ and allows 70% of the cells to survive. Additionally, just as the reviewer suggests, from our CRISPR screen, multiple genome maintenance mechanisms other than TLS contribute to tolerance of TMZ genotoxicity. For instance, in TLS-deficient cells ssDNA gaps undergo nucleolytic processing to generate DSB. Therefore, NHEJ and HR are back-up pathways that contribute to repair of DSB that arise in TLS-deficient cells. In fact we published a similar role for NHEJ in repair of DSB that arise secondarily from ROS-induced ssDNA in RAD18^{-/-} cells (PMID: 23295675).

Lines 216-217. Here the authors look at RAD18- POLK- double mutants. Again, with the small figure and y-axis log scale it is difficult to say that POLK- phenocopies RAD18- because all have different level of survival.

Thank you – we agree. In the revised manuscript we describe this as “partially phenocopied” (line 225). It is established that RAD18 participates in the HR pathway independently of its TLS functions (PMID: 19396164), while POLK can be activated independently of RAD18 (PMID: 16308320). Because *RAD18* and *POLK* are not invariably epistatic we would not predict that POLK-loss would fully phenocopy RAD18-deficiency.

Line 256, FA is not defined.

Thank you – we have now defined the ‘Fanconi Anemia’ (FA) pathway (line 266).

Line 323. If RAD18+ is involved in error-prone bypass of N7mG and N3mA, is this using pol zeta? If this is MMR independent, what triggers RAD18 here and since all three types of lesions happen after TMZ treatment, once RAD18 is activated could it not act on O6mG?

This is a very good question that is not directly addressed by our study. Future mutagenesis studies with cell lines lacking individual DNA polymerases will be necessary to identify the enzyme(s) that mediate error-free and mutagenic bypass of the different TMZ-induced lesions. However we agree that involvement of Pol zeta as an extender is highly likely. Pena-Diaz et al demonstrated (PMID: 22864113) that Pol eta is recruited to chromatin following alkylation damage. Therefore we hypothesize that Pol eta inserter activity mediates the error-prone bypass of N7mG and N3mA, perhaps with participation of Pol zeta as an extender.

Line 371-372. Here the authors suddenly start discussing other DNA damage tolerance and binding of polK vs polH to PCNA. This is relevant but should have been introduced at the beginning. And the novel role, undefined of RAD18 in error-prone (not error-free) bypass of N7mG and N3mA (line 414) seems to come out of nowhere with no molecular explanation or hint of the other role.

Thank you - we agree completely that the relationship between Polk and Polh needs some additional context, particularly for readers not familiar with the TLS field. Therefore, in the revised manuscript we have described more clearly why competition between Pol kappa and Pol eta is relevant and explains our results (lines 379-382).

Again, we thank all three reviewers for urging us to explain the role of RAD18 in error-prone (not error-free) bypass of N7mG and N3mA. In the revised manuscript (lines 429-432) we describe how the mechanism of RAD18- and MMR-dependent mutagenesis (T>A, C>A, T>C, C>G and T>G) caused by N7mG and N3mA is explained by previous findings of Pena-Diaz et al. Those workers demonstrated that DNA replication-independent noncanonical mismatch repair (ncMMR) recruits error-prone TLS polymerases to bypass alkylating DNA lesions.

Reviewer #3 (Remarks to the Author):

In this manuscript, the authors reported the activation RAD18 as part of the TMZ-induced DDR in GBM cells, which mediates the repair and tolerance of O6mG lesions through Y-family DNA polymerase POLK. Moreover, they performed a CRISPR KO screening in both RAD18 wildtype and KO cells and provided a DDR landscape by which GBM cells resist TMZ-induced genotoxicity. Lastly, they determined the impact of RAD18 on TMZ-induced mutagenesis. This is a potentially interesting study with possible significance. Although the novelty is dampened by a recent publication, Hanisch et al, Cell Death & Disease, 2022, which reported the activation and function of RAD18 in TMZ-treated glioma cells, their unbiased CRISPR screening provide a better understanding of DDR networks after TMZ treatment. Overall, the manuscript is well written with clear presentation, and quality of data is acceptable. The conclusions are justified and of interest to the diverse audience of Nature Communications. The authors may consider the following comments to strengthen the manuscript.

1. The authors should confirm their major conclusions in animal models, such as the impact of RAD18-KO on TMZ sensitivity and the synergetic effect of RAD18-KO and POLD3 KO on TMZ sensitivity.

We agree completely that the logical next phase of this project will involve mouse experiments. However, we consider preclinical work to be well beyond the scope of this paper. As another reviewer already commented, there are already “numerous types of experiments and large numbers of data in this paper” which might jeopardize clarity and flow. We have already generated an unprecedented and comprehensive atlas of how the glioblastoma DDR responds to therapy-induced genotoxicity to dictate viability and mutability outcomes. Moreover, even although the focus of this study is on mechanism, we have validated our findings in a novel organotype brain slice culture system using glioblastoma cells growing in their pathological setting (Fig. 2H).

2. GBM cells with different genetic background might rely on difference DNA repair pathway to survive TMZ

treatment when RAD18 is missing. In Fig. 5, the authors performed all experiments in only one GBM cell lines, U373. It is recommended that the authors repeat some of the synergetic experiments in some other GBM lines or patient-derived glioma stem-like cells.

We agree completely. In the revised manuscript we have repeated some of the key experiments and validated results of POLD3 and Chk2 ablation studies using an additional U87 GBM line (Supplementary figure 6C-D).

3. The activation and function of RAD18 had been reported by Hanisch et al, Cell Death & Disease, 2022. The authors should cite this paper and discuss their study.

Thank you for drawing our attention to this nice study. We have cited and discussed the findings of Hanisch et al. in our revised manuscript (lines 387-387).

4. The authors' conclusion from Fig. 4B and 4H regarding to the effect of POLH knockdown on TMZ sensitivity is contradictory. Is it caused by different TMZ dose? The authors should perform the experiment in Fig. 4B with higher TMZ dose.

Yes, just as the reviewer suggests the effect of Pol eta knockdown depends on the TMZ dose. In the revised manuscript we included all the data from the experiment in Fig. 4B which was originally performed using two doses of TMZ. As shown in the new Fig. 4B, PolH knockdown causes a statistically significant increase in survival at 4 micromolar TMZ, fully consistent with our other results.

5. In Fig 5D and E, in addition to showing $-\log_{10}$ P value, the authors should also show the fold change value. Moreover, all abbreviations should be described in figure legend.

We agree. In the revised manuscript, we present the data as heat maps and radar plots with both $-\log_{10}$ p-value and \log_2 fold-change.

6. In line 279, "Fig. 5H" should be changed to "Fig. 5I".

Thank you for noticing this mistake. We have corrected this in the revised manuscript.

7. The numbers in Fig 6B and description in line 303-305 is not matched. Is the y-axis of Fig 6B correctly labeled? Does it mean the number of total SNV or the number of read counts of SNV? They should show the number of total SNV rather than read counts.

Apologies for not explaining clearly: The SNV counts referred to in the text indicate the TMZ-induced component. i.e. We subtracted the basal SNV counts (untreated cells) from the SNV counts in TMZ-treated cells. In the revised manuscript we explicitly state that we are referring to the TMZ-induced component of the SNV.

8. The authors should provide more detailed information of all the patient samples they used for analysis in Fig. 7. Is it recommended to provide a table including the clinical features, genetic background, and previous treatment strategies for each patient.

We agree and have included the patient clinical data used for our analysis in Supplemental Table 8.

Reviewers' Comments:

Reviewer #1:

Remarks to the Author:

The revised manuscript from Cheng et al., is significantly improved and addresses the major concerns.

Minor comments:

- Figure legend for Figure 1D indicates the scale bar as 10 μ M when they mean μ m.
- The graph key for Figure 3C has a type-o in the word Control
- The flow cytometry data in Figure 3C are not labeled for WT and RAD18 deficient populations
- Although not essential to the main storyline, it seems like Rad18 knockdown in Figure 4G leads to redistribution of polK outside of the nucleus. Is this a consistent finding or just an artefact?

Reviewer #2:

Remarks to the Author:

The authors have taken all the reviewers' comments seriously and performed additional experiments. These make the results more compelling and provide a new understanding of mutation and resistance in TMZ-treated GBM.

Reviewer #3:

Remarks to the Author:

The authors addressed most of my comments except the following two.

1. One is to "confirm their major conclusions in animal models." Although the authors presented strong arguments for not doing the in vivo studies, but the clinical relevance is markedly diminished without the animal model. In fact, U373 glioma cell line has been considered as a "contaminated or identity in questions cell line" with U251 and SNB19 cell lines that has been described by ATCC <https://www.atcc.org/the-science/authentication/reclassified-cell-lines> Other source that also showed the concern for U373 cell line

<https://www.culturecollections.org.uk/services/authenticcell/misidentifiedcelllines.aspx>

https://www.cellosaurus.org/CVCL_2219

Since 95-98% data of this study was done in U373 cell line, confirmation of the major finding in this study in an orthotopic glioma brain tumor xenograft model is a must.

2. The second cell line U87 used in Fig. 6C-D suffers the similar criticism as this cell line does not reflect clinically relevant phenotype in the brain of mice. Thus, the authors must repeat the key experiments in a PATIENT-DERIVED glioma cells or glioma stem-like cell line or a patient-derived xenografted GBM cells (GBM PDX). This is a common practice in all studies are currently on going in neuro-oncology field.

We would like to submit our revised manuscript NCOMMS-22-50932A entitled: "*Trans-Lesion Synthesis and Mismatch Repair Pathway Crosstalk Defines Chemoresistance and Hypermutation Mechanisms in Glioblastoma*" for publication in Nature Communications.

This is a second revision. Our point-by-point responses to the most recent reviewer comments are detailed below.

I do hope this manuscript will be acceptable.

Reviewer 1

Major comments:

"The revised manuscript from Cheng et al., is significantly improved and addresses the major concerns."

Minor comments:

"Figure legend for Figure 1D indicates the scale bar as 10 μ M when they mean μ m; The graph key for Figure 3C has a type-o in the word Control; The flow cytometry data in Figure 3C are not labeled for WT and RAD18 deficient populations; Although not essential to the main storyline, it seems like Rad18 knockdown in Figure 4G leads to redistribution of polk outside of the nucleus. Is this a consistent finding or just an artefact?"

Response:

We have corrected the minor typos and errors with annotation of figures. Regarding the redistribution of Polk to outside the nucleus in RAD18-depleted cells - no, this is not artifact. This observation is related our previous discovery that RAD18 is a molecular chaperone that directs TLS polymerases to sites of replication stalling. We reported this in our 2010 JCB paper (PMID: 21098111). We agree completely with the reviewer that this observation is tangential to the main thrust of this report and we have not commented on this incidental result in the manuscript. We thank Reviewer 1 for all the helpful suggestions for improvement of our manuscript.

Reviewer 2

Comments:

"The authors have taken all the reviewers' comments seriously and performed additional experiments. These make the results more compelling and provide a new understanding of mutation and resistance in TMZ-treated GBM."

Response:

We are delighted that the reviewer finds this work compelling and appreciates that this study contributes significantly to our understanding of mutation and resistance in TMZ-treated GBM. We thank our colleague for all their help and support with this study.

Reviewer 3

Major Comments.

1. One is to "confirm their major conclusions in animal models." Although the authors presented strong arguments for not doing the in vivo studies, but the clinical relevance is markedly diminished without the animal model. In fact, U373 glioma cell line has been considered as a "contaminated or identity in questions cell line" with U251 and SNB19 cell lines that has been described by ATCC <https://www.atcc.org/the-science/authentication/reclassified-cell-lines> Other source that also

showed the concern for U373 cell line

<https://www.culturecollections.org.uk/services/authenticcell/misidentifiedcelllines.aspx>

https://www.cellosaurus.org/CVCL_2219

Since 95-98% data of this study was done in U373 cell line, confirmation of the major finding in this study in an orthotopic glioma brain tumor xenograft model is a must.

Response regarding the use of animal models:

(I) We have provided an unparalleled mechanistic and genetic analysis of the genome maintenance mechanisms that process TMZ-induced genotoxicity.

(II) The study is already *very* comprehensive. Not only did we perform a careful biochemical dissection of the TLS pathway and genetic screening to define the broader TMZ-responsive DDR landscape; we also characterized effects of TMZ and DNA repair status on the mutational landscape. Nobody has ever performed such a comprehensive analysis of the DDR response to TMZ and its impact on genome stability in GBM (or any setting). Our current findings are highly significant and impactful.

(III) The manuscript has already been approved by two reviewers. One of those reviewers originally commented that this manuscript already has too much data. Now Reviewer #3 is requesting additional mouse work.

(IV) As an innovative alternative to mouse work we have performed key experiments using a newly-developed organotype brain-slice culture platform (Cell Rep Med 2023 Jun 20;4(6):101042. A living ex vivo platform for functional, personalized brain cancer diagnosis. PMID: 37192626)

(V) This is already a complete stand-alone DNA repair study with ample new mechanistic information. Although our findings have obvious therapeutic implications, this was never intended to be a preclinical study – which is what Reviewer #3 is requesting. Frankly it is both remarkable and disappointing to us that Reviewer #3 has no appreciation for the new mechanisms and paradigms generated by our study.

Response regarding the use of the U373 cell line.

I object very strongly to the misleading comments of Reviewer #3 regarding the U373 cell line. The reviewer is distorting facts regarding this GBM line.

Please look at the ATCC link that s/he sent: <https://www.atcc.org/the-science/authentication/reclassified-cell-lines>

The non-issue which the reviewer is trying to exaggerate is that some clones of the U373 GBM line distributed to ATCC were mixed up with a different GBM line (U251). There is not any problem with U373 cells. We have confirmed the identity of our U373 clones by finger-printing ('Materials and Methods'). However, even if the U373 cells we used were actually U251 cells *this would not affect any of our conclusions in the slightest*. Our use of U373 cells in no way detracts from our findings concerning DDR mechanisms and how TMZ causes genome stability. U373 cells are used extensively by many reputable neuro-oncology researchers, including some of our close UNC colleagues who developed the elegant organotype brain slice culture platform described in our manuscript. In fact, our colleagues used U373 cells to validate their novel personalized medicine platform (Cell Rep Med 2023 Jun 20;4(6):101042. *A living ex vivo platform for functional, personalized brain cancer diagnosis*. PMID: 37192626).

"2. The second cell line U87 used in Fig. 6C-D suffers the similar criticism as this cell line does not reflect clinically relevant phenotype in the brain of mice. Thus, the authors must repeat the key experiments in a PATIENT-DERIVED glioma cells or glioma stem-like cell line or a patient-derived xenografted GBM cells (GBM PDX). This is a common practice in all studies are currently on going in neuro-oncology field."

Response regarding validation of key experiments in cell line. We agree it is useful to determine the extent to which genome maintenance mechanisms are generalizable between cancer cell lines, and primary patient-derived tumors. Therefore, we have collected a panel of GBM cells (including PDX) and we have now validated all our key findings in multiple GBM settings. Those results are presented in Supplementary Figures 7,8 and 9 in our revised manuscript. The GBM models we have used for these experiments, and our key findings are described below.

(I) GBM models used in the revised manuscript include: LN229 and D54 established GBM cell lines; MS21 a low-passage patient-derived line (generated by our UNC colleague Dr. Hingtgen and described in Mann, B. et al 2023), GBM8 (a Glioblastoma Stem Cell or GSC-enriched line); and several GBM PDX lines from the Mayo clinic: GBM12 (a GSC line), GBM75, GBM85 (a GSC line), and GBM123.

(II) We demonstrate TMZ-inducible TLS pathway activation (using PCNA-mono-ubiquitination as a TLS marker) in all MGMT-deficient cell lines (LN229, D54, MS21, GBM8, GBM12, GBM75, GBM85). As expected, TMZ-induced TLS activation was undetectable in MGMT-proficient line GBM123.

(III) Using gene editing to delete MLH1 (MMR factor) we verified that TMZ-induced TLS activation is MMR-mediated in GBM cell lines LN229 and D54. Using siRNA to ablate RAD18 and MSH2 (MMR factor), we further confirmed that TMZ-induced TLS pathway activation is MMR-dependent in the patient-derived GBM line MS21 and the PDX line GBM75.

(IV) We show that RAD18-deficiency sensitizes LN229, D54, MS21 and GBM75 to TMZ.

(V) We show that TMZ and DNA-PK inhibitor treatments have a higher synergy score in *RAD18*^{-/-} cells (generated by CRISPR) when compared with parental *RAD18*^{+/+} LN229 and D54 GBM (fully reproducing data we generated in U373 cells).

(VI) We also performed experiments to test the effect of RAD18-deficiency in GSC-enriched lines. We obtained a very interesting result, namely that RAD18 was necessary for maintaining cancer stemness features (even in the absence of chemotherapy treatment). This result indicates that TLS is important for GBM stem cells to tolerate intrinsic stresses, and that RAD18 confers at least two tumorigenic properties: cancer stemness and chemoresistance. However, because the focus of this manuscript is on responses to TMZ, we have omitted the results of our studies with GSC. We mention this here solely to indicate that we have made a good-faith effort to address the reviewer comments.

Reviewers' Comments:

Reviewer #4:

Remarks to the Author:

In this work, the authors have dissected the role of RAD18, TLS polymerases and MMR in the response of GBM cells to TMZ and in hypermutation.

Glioma stem-like cells (GSCs) have been proposed to foster resistance to DNA-damaging treatment, in part through constitutive activation of the DNA damage response (see e.g. *Cancer Res* (2018) 78 (17): 5060–5071. <https://doi.org/10.1158/0008-5472.CAN-18-0569>), which makes them particularly relevant for the present study. However, although most of the work was done with “GBM” cell lines grown in serum-containing medium (which can alter their genomes/transcriptomes, as well as DDR and RS response), I recognize that the authors have included experiments using GSCs and other relevant models to validate their important findings satisfactorily.

Thus, although I agree with Reviewer #3 that validation of the major findings in an orthotopic xenograft model of GBM would raise its clinical relevance significantly, I think the current manuscript, which represents an important contribution to our knowledge of the mechanisms underlying chemoresistance and hypermutation in glioblastoma, is of sufficient scientific interest to justify its publication in *Nature Communications*.

I would like to raise the following points that need to be addressed:

Major points

1. In the Discussion, the authors state “Here we show coordinate activation of CHK1 and RAD18 in response to TMZ-induced DNA damage, consistent with a common RPA/ssDNA-based activation mechanism for both ATR/CHK1 and RAD18 pathways.” Indeed, in Lane 127 and in subsequent sections, the authors used P-CHK1 as a measure of ATR activation in TMZ-treated cells. However, in relevant instances, CHK1 can also be phosphorylated by other DDR kinases such as ATM (*J Biol Chem*. 2003 Apr 25;278(17):14806-11. doi: 10.1074/jbc.M210862200.) or DNA-PK (Buisson, R., et al. (2015). *Mol. Cell* 59, 1011–1024. doi: 10.1016/j.molcel.2015.07.029). It would therefore be important to provide controls where ATR is inhibited, to verify that P-CHK1 is indeed mediated by ATR and thus better establish the link with the accumulation of RPA-coated ssDNA. This is especially important as the authors i) show that NHEJ is an important pathway in RAD18^{-/-} cells and ii) they later validate the importance of DNA-PK inhibition in TMZ sensitization. Likewise, in most of the Figures, the authors show P-ATM but do not address whether P-CHK1 and P-CHK2 are mediated by ATM or ATR (or DNA-PK, see above). Specific inhibitors should be used to identify which apical kinases mediate such phosphorylations.

2. For the sake of clarity, the authors should incorporate the outcome (e.g. life/death) of inactivating the replication stress factors studied in this work (together with MGMT expression), on the “hypermutation” model of Figure 7B.

Other points:

3. Throughout the manuscript, the authors should add more references to justify the use of specific DDR markers in their experiments.

4. Figure 2A (lane 1081-1082): to what level of cytotoxicity does 100 microM TMZ correspond for NHA and NHARas cells?

5. Lane 189: Define scRNA

6. Figure 3C: The authors should label the 4 windows of the upper panel so we know exactly to what cell lines they correspond.

7. Figure 3H: The PCNA and PCNA-Ub labels are inverted.
8. Lane 214-223: The authors should specify which cell line was used for Fig. 4B. Furthermore, they indicate that REV1 inhibition does not sensitize to TMZ. However, unlike for polH depletion, they do not comment on the fact that this inhibition actually results in significant resistance to TMZ in U373 (Figure 4B, rightmost panel).
9. Figure 4C middle panel: there needs to be a space between "PolK" and "foci" in the Y-axis label.
10. Figure 5A: Define BSD.
11. Figure 5A: isn't it "Shield" instead of scheild ? The authors should define all the terms used in this screen, as well as detail their relevance for the screen.
12. Lane 282: To facilitate the reading, and before their validation, the authors should mention that POLD3 etc are some of the relevant hits of their screen.
13. Lane 295-299: The MGMT methylation status of all cell lines should be mentioned in the text.
14. Supplementary Figure 7: the authors use 10 microM TMZ (A) and 100 microM TMZ (B) in vitro. These are massive doses considering that their clonogenicity assay (C), done at a TMZ range between 0 and 2.5 microM, shows that the viability of WT and RAD18-/- LN229 cells is crucially affected already at such small doses. What observations on DNA damage and DDR markers are made when smaller doses are used (e.g., corresponding to the IC20)?
15. In the text, for the data shown in Supp Figures 7A, 7G, 8A, 8B, 8F, 8G, 9A and 9E, the authors should reference the ubiquitination of gH2AX and detail why they have examined this modification.

REVIEWERS' COMMENTS

Reviewer #4 (Remarks to the Author):

In this work, the authors have dissected the role of RAD18, TLS polymerases and MMR in the response of GBM cells to TMZ and in hypermutation.

Glioma stem-like cells (GSCs) have been proposed to foster resistance to DNA-damaging treatment, in part through constitutive activation of the DNA damage response (see e.g. Cancer Res (2018) 78 (17): 5060–5071. <https://doi.org/10.1158/0008-5472.CAN-18-0569>), which makes them particularly relevant for the present study. However, although most of the work was done with “GBM” cell lines grown in serum-containing medium (which can alter their genomes/transcriptomes, as well as DDR and RS response), I recognize that the authors have included experiments using GSCs and other relevant models to validate their important findings satisfactorily.

Thus, although I agree with Reviewer #3 that validation of the major findings in an orthotopic xenograft model of GBM would raise its clinical relevance significantly, I think the current manuscript, which represents an important contribution to our knowledge of the mechanisms underlying chemoresistance and hypermutation in glioblastoma, is of sufficient scientific interest to justify its publication in Nature Communications.

We would like to thank your reviewer for the kind comments on our manuscript (NCOMMS-22-50932B).

We are delighted that the reviewer realizes that our experiments “*validating our important findings*” in Cancer Stem Cells (CSC) makes the results even more relevant.

We are also pleased that the reviewer deems our study to be “*of sufficient scientific interest to justify its publication in Nature Communications.*”

We appreciate the reviewer's helpful suggestions for improvement. We have revised our manuscript to incorporate those suggestions, as detailed in our point-by-point response below.

We hope our revised manuscript will be acceptable for publication.

Major points

1. In the Discussion, the authors state “Here we show coordinate activation of CHK1 and RAD18 in response

to TMZ-induced DNA damage, consistent with a common RPA/ssDNA-based activation mechanism for both ATR/CHK1 and RAD18 pathways.” Indeed, in Lane 127 and in subsequent sections, the authors used P-CHK1 as a measure of ATR activation in TMZ-treated cells. However, in relevant instances, CHK1 can also be phosphorylated by other DDR kinases such as ATM (J Biol Chem. 2003 Apr 25;278(17):14806-11. doi: 10.1074/jbc.M210862200.) or DNA-PK (Buisson, R., et al. (2015). Mol. Cell 59, 1011–1024. doi: 10.1016/j.molcel.2015.07.029). It would therefore be important to provide controls where ATR is inhibited, to verify that P-CHK1 is indeed mediated by ATR and thus better establish the link with the accumulation of RPA-coated ssDNA. This is especially important as the authors i) show that NHEJ is an important pathway in RAD18-/- cells and ii) they later validate the importance of DNA-PK inhibition in TMZ sensitization. Likewise, in most of the Figures, the authors show P-ATM but do not address whether P-CHK1 and P-CHK2 are mediated by ATM or ATR (or DNA-PK, see above). Specific inhibitors should be used to identify which apical kinases mediate such phosphorylations.

The reviewer is absolutely correct that there is often redundancy between the ATM and ATR kinases and that there is often compensatory activation of CHK1 by ATM when ATR is inhibited (and reciprocally compensatory

CHK2 activation by ATR when ATM is inhibited). We feel very strongly however that there will be no added value to performing ATM/ATR inhibitor treatments, in part for the very reason the reviewer mentions: it is well known that there is dynamic rewiring of these checkpoint pathways and redundancy between ATM and ATR for activating CHK2 and CHK1 respectively. *Whether ATM, ATR, or both kinases contribute to TMZ-induced CHK1/CHK2 phosphorylation, and the extent to which these apical kinases compensate for each other in our experiments has absolutely no impact on any of our conclusions.* Our major point here is that RAD18 activation and PCNA ubiquitination are well-established consequences of RPA-ssDNA accumulation. There is no better explanation for the TMZ-induced RAD18-PCNA responses we observe. It is true that DSB can activate ATM, then generate ssDNA (via DSB resection) that activate CHK1 activation secondarily and indirectly. However, RAD18 activation and PCNA ubiquitination do not occur in response to DSB resection.

We believe the best way to address this issue is to explain that *"it is formally possible that the apical kinases ATM and ATR both contribute to TMZ-induced CHK1/CHK2 phosphorylation. However, RAD18-dependent PCNA mono-ubiquitination is best explained by TMZ-induced accumulation of ssDNA independently of DSB formation, as we have shown previously for other genotoxins."* This has been stated in the revised manuscript (lines 131-134).

With respect, it is unclear to us exactly why the reviewer feels *"this is especially important as the authors i) show that NHEJ is an important pathway in RAD18-/- cells and ii) they later validate the importance of DNA-PK inhibition in TMZ sensitization."* We first reported that in RAD18/TLS-defective cells, ssDNA is aberrantly processed to DSB (Bi et al., 2005, PMID: 15817457; 2006 PMID: 16611994). This is now an established hallmark of TLS-deficiency that many others have subsequently observed. We have also previously published that RAD18/TLS-compromised cells rely on NHEJ for remediation of DSB (Yang et al., 2013 PMID: 23295675). This mechanism is established and completely uncontroversial. I believe we failed to explain these issues in the original manuscript. Therefore, in the revised manuscript we have provided additional citations of past work and explained the relationship between RAD18/TLS-deficiency, DSB, and DSB repair more appropriately (lines 131-134).

2. For the sake of clarity, the authors should incorporate the outcome (e.g. life/death) of inactivating the replication stress factors studied in this work (together with MGMT expression), on the "hypermutation" model of Figure 7B.

We thank the reviewer for the excellent idea. In the revised manuscript we have incorporated the survival/death outcomes into Figure 7B as suggested.

Other points:

3. Throughout the manuscript, the authors should add more references to justify the use of specific DDR markers in their experiments.

Thank you for the suggestion. We agree and have added citations explaining the use of various DDR markers throughout the results sections, e.g. line 128 (describing pCHK1 and pATM as markers of ssDNA and DSB); line 164 - describing pRPA32 (a ssDNA marker), pATM, and γH2AX (DSB markers, and elsewhere).

4. Figure 2A (lane 1081-1082): to what level of cytotoxicity does 100 microM TMZ correspond for NHA and NHARas cells?

The TMZ dose-response curves in Fig. 2B of the revised manuscript show that 100μM TMZ is sub-lethal for NHA and lethal to NHARas cells.

5. Lane 189: Define scRNA

Apologies for using the abbreviation scRNA. In the revised manuscript we refer more appropriately to "single cell RNA". (line 193)

6. Figure 3C: The authors should label the 4 windows of the upper panel so we know exactly to what cell lines they correspond.

Thank you for the suggestion. In the revised manuscript we properly describe and label these as representative images of U373 WT cells (see Fig. 3c and the accompanying legend).

7. Figure 3H: The PCNA and PCNA-Ub labels are inverted.

Thank you very much for noticing this mistake. We have correctly labeled Fig. 3h in the revised manuscript.

8. Lane 214-223: The authors should specify which cell line was used for Fig. 4B. Furthermore, they indicate that REV1 inhibition does not sensitize to TMZ. However, unlike for polH depletion, they do not comment on the fact that this inhibition actually results in significant resistance to TMZ in U373 (Figure 4B, rightmost panel).

Thank you for the suggestion. In the revised manuscript we specify that U373 cells were used for Fig. 4B.

As the reviewer notes, the effects of POLH ablation on survival are phenocopied by REV1 inhibition. We have already explained why POLH knockdown relieves competition for POLK. POLH and REV1 associate with each other and are epistatic for TLS, explaining why REV1 inhibition recapitulates the effects of POLH deficiency. This is explained in the revised manuscript (line 396).

9. Figure 4C middle panel: there needs to be a space between “PolK” and “foci” in the Y-axis label.

Thank you for noticing this mistake which is corrected in Fig. 4c of the revised manuscript.

10. Figure 5A: Define BSD.

In the revised manuscript we define Blasticidin (BSD) in the 'Methods' section (line 688).

11. Figure 5A: isn't it “Shield” instead of scheild ? The authors should define all the terms used in this screen, as well as detail their relevance for the screen.

Thank you for noticing this typographical error. Shield-1 is the compound used to stabilize the destabilizing domain (DD) of Cas9 protein. In the revised manuscript we have defined Shield-1 and explained why it was used in the CRISPR screen (line 729).

12. Lane 282: To facilitate the reading, and before their validation, the authors should mention that POLD3 etc are some of the relevant hits of their screen.

Thank you for the suggestion. We have modified the relevant text (line 286 of the revised manuscript) as recommended.

13. Lane 295-299: The MGMT methylation status of all cell lines should be mentioned in the text.

We agree this is important information. The MGMT methylation status of all cell lines is now described in the revised manuscript (lines 300-303).

14. Supplementary Figure 7: the authors use 10 microM TMZ (A) and 100 microM TMZ (B) in vitro. These are massive doses considering that their clonogenicity assay (C), done at a TMZ range between 0 and 2.5 microM, shows that the viability of WT and RAD18-/- LN229 cells is crucially affected already at such small doses. What observations on DNA damage and DDR markers are made when smaller doses are used (e.g., corresponding to the IC20)?

In our experiments, the induction of DDR markers at TMZ doses corresponding to the IC20 was very modest and sometimes undetectable. However, it is very well accepted in the DNA repair field that the concentrations

of genotoxins required to elicit a detectable DDR signal (e.g. pCHK1, PCNA-Ub etc) exceed the concentrations necessary to elicit phenotypic effects such as lethality. This is due to limitations in the sensitivity of the readouts used to detect the markers of DNA replication fork stalling and DDR activity (such as pCHK1 and PCNA-Ub antibodies or TLS polymerase foci). For example, the dependency on TLS-deficiency for proper S-phase checkpoint recovery from is very evident at 50-100 nM BPDE (a replication fork-stalling genotoxic carcinogen). However, 600 nM BPDE is required to induce a detectable level of PCNA mono-ubiquitination and TLS polymerase recruitment to replication forks (PMID: 16611994). In other words, to detect DDR signals triggered by replication fork stalling a threshold number of stalled DNA replication forks is required. Attaining this threshold for detection requires higher levels of DNA replication fork-stalling DNA lesions than is needed to kill the cells. Please note too that the doses of TMZ used in our study are in the same range as those other investigators have used to study TMZ-induced DDR signaling. For example Bandey et al. used 250-350 micromolar TMZ for measuring gH2AX responses (PMID24793792), Aasland et al., used 100 micromolar TMZ to induce p53 phosphorylation (PMID3036124), and Agnihotri et al., used 100 micromolar TMZ to detect activation-associated ATM phosphorylation (PMID 25100205). These points have been made in the revised manuscript (see note at end of legend for Supplementary Figure 7).

15. In the text, for the data shown in Supp Figures 7A, 7G, 8A, 8B, 8F, 8G, 9A and 9E, the authors should reference the ubiquitination of gH2AX and detail why they have examined this modification.

Apologies for not explaining why we showed the mono-ubiquitinated gH2AX bands on those immunoblots. In many neoplastic cells, the signals for both basal and genotoxin-induced gH2AX on immunoblots are strong, easily saturated and exceed the dynamic range, precluding meaningful quantitative comparisons between experimental samples. However, in response to spontaneous or genotoxin-induced DNA damage, fold changes in levels of mono-ubiquitinated gH2AX track closely with fold changes in non-ubiquitinated gH2AX and also provide a good surrogate marker for DSB. Mono-ubiquitinated gH2AX levels are typically far less abundant than those of gH2AX and do not generate saturated signals. Thus differences in levels of mono-ubiquitinated gH2AX between samples can be determined and quantified more appropriately. These points have been noted in the legend to Supplementary Fig. 7.